# The budding yeast Fkh1 Forkhead associated (FHA) domain promotes a G1-chromatin state and the activity of chromosomal DNA replication origins

Timothy Hoggard[1], Erika Chacin[2], Allison J. Hollatz[1,3], Christoph F. Kurat🄳[2], Catherine A. Fox🄳[1,3]*

1 Department of Biomolecular Chemistry, School of Medicine and Public Health, University of Wisconsin, Madison, Wisconsin, United States of America, 2 Biomedical Center Munich (BMC), Division of Molecular Biology, Faculty of Medicine, Ludwig-Maximilians-Universität in Munich, Martinsried, Germany, 3 Integrated Program in Biochemistry, University of Wisconsin, Madison, Wisconsin, United States of America

* cfox@wisc.edu

**Data Availability Statement:** ORC ChIPSeq BioProject PRJNA694026; Fkh1 ChIPchip GEO

## Abstract

In *Saccharomyces cerevisiae*, the forkhead (Fkh) transcription factor Fkh1 (forkhead homolog) enhances the activity of many DNA replication origins that act in early S-phase (early origins). Current models posit that Fkh1 acts directly to promote these origins' activity by binding to origin-adjacent Fkh1 binding sites (FKH sites). However, the post-DNA binding functions that Fkh1 uses to promote early origin activity are poorly understood. Fkh1 contains a conserved FHA (forkhead associated) domain, a protein-binding module with specificity for phosphothreonine (pT)-containing partner proteins. At a small subset of yeast origins, the Fkh1-FHA domain enhances the ORC (origin recognition complex)-origin binding step, the G1-phase event that initiates the origin cycle. However, the importance of the Fkh1-FHA domain to either chromosomal replication or ORC-origin interactions at genome scale is unclear. Here, S-phase SortSeq experiments were used to compare genome replication in proliferating *FKH1* and *fkh1-R80A* mutant cells. The Fkh1-FHA domain promoted the activity of ≈ 100 origins that act in early to mid- S-phase, including the majority of centromere-associated origins, while simultaneously inhibiting ≈ 100 late origins. Thus, in the absence of a functional Fkh1-FHA domain, the temporal landscape of the yeast genome was flattened. Origins are associated with a positioned nucleosome array that frames a nucleosome depleted region (NDR) over the origin, and ORC-origin binding is necessary but not sufficient for this chromatin organization. To ask whether the Fkh1-FHA domain had an impact on this chromatin architecture at origins, ORC ChIPSeq data generated from proliferating cells and MNaseSeq data generated from G1-arrested and proliferating cell populations were assessed. Origin groups that were differentially regulated by the Fkh1-FHA domain were characterized by distinct effects of this domain on ORC-origin binding and G1-phase chromatin. Thus, the Fkh1-FHA domain controlled the distinct chromatin architecture at early origins in G1-phase and regulated origin activity in S-phase.

GSE165464; Sequencing files for SortSeq and MNaseSeq BioProject PRJNA1076901.

**Funding:** This work was supported by the NIH (R35GM141641 to CAF), including salary support for TH, AJH and CAF, and by the Deutsche Forschungsgemeinschaft (DFG) (the German Research Foundation) (project ID 213249687—SFB 1064 to CFK). The funders had no role in study design, data collection and analysis, decision to publish, or preparation of the manuscript.

**Competing interests:** The authors have declared that no competing interests exist.

## Author summary

DNA replication must be regulated both spatially and temporally to insure the accurate and efficient duplication of the eukaryotic genome. Altering this spatiotemporal control can cause mistakes in genome copying and/or deficiencies in cell proliferation that promote disease. Therefore, the proteins and mechanisms underlying the normal spatiotemporal progression of eukaryotic genome duplication are of keen interest. The Fkh1 protein, a type of DNA binding protein that regulates eukaryotic cell proliferation, contributes to the spatiotemporal control of genome duplication in budding yeast. We learned that a single amino acid change within one region of the Fkh1 protein, named the FHA domain, altered the spatiotemporal progression of budding yeast genome duplication. FHA domains convey molecular information by directly binding to partner proteins that are phosphorylated on threonine residues. This information in turn stimulates specific molecular activities or events required by the cell. Thus, our study revealed that a protein-protein interaction controlled by threonine-phosphorylation is required for the normal spatiotemporal progression of yeast genome duplication.

## Introduction

Efficient and accurate duplication of the eukaryotic genome requires that multiple independent DNA replication origins, the loci that initiate DNA replication during S-phase, are distributed both spatially and temporally across each chromosome. Biochemical and structural progress have provided the field with a clear picture of the core proteins and steps required to form an origin [1]. Briefly, the origin cycle can be divided into two cell-cycle restricted phases. First, in G1-phase, the ORC (origin recognition complex) and the Cdc6 protein form a complex on the DNA that directs the loading of the DNA replicative helicases in an inactive form called the MCM (mini chromosome maintenance) complex (origin licensing phase) (reviewed in [2]). Second, in S-phase, additional origin-control proteins associate with the inactive MCM complex and promote its remodeling into two active helicases that bidirectionally unwind DNA (origin activation phase). These steps occur at every DNA locus that acts as an origin. However, the local chromatin composition of the locus has an impact on the probability that either the G1-phase licensing and/or S-phase activation reactions will proceed to completion (reviewed in [3]). This chromatin-influenced variation in origin-reaction probabilities contributes to the stochasticity of origin-use, such that each individual S-phase uses a distinct cohort of origins to replicate the cellular genome. Another result is the generation of a characteristic spatial and temporal pattern of genome replication that can be observed at the cell population level and that is linked to both genome stability and cell identity [4]. While the essential mechanics of the origin cycle are now understood in molecular detail, the mechanisms by which local origin-adjacent chromatin alters origin activity are not.

In *Saccharomyces cerevisiae* (budding yeast), the forkhead transcription factors (Fkhs), specifically the paralogs encoded by *FKH1* and *FKH2*, have emerged as important non-histone chromatin regulators of this organism's DNA replication origins [5]. In contrast to the core origin-control proteins, the Fkhs are not essential for cell division [6]. Instead, Fkhs1/2 act in a partially redundant manner to promote cell-cycle regulated transcription and to enhance the activity of early S-phase origins (referred to as Fkh1/2-activated origins) [7–10]. The Fkh proteins are posited to promote the activity of many early origins within the yeast genome via a direct mechanism, requiring that Fkhs bind to origin-adjacent Fkh binding sites (FKH sites).

Thus, the origin-regulatory roles of the Fkh proteins are viewed as distinct from and not an indirect consequence of their roles in gene transcription [10–14]. Multiple non-mutually exclusive mechanisms have been proposed for how Fkhs promote early origin activity. The most definitive molecular explanation to date is that the Fkh proteins physically interact with the essential S-phase kinase DDK (Dbf4 dependent kinase) that is present in limiting levels relative to licensed origins, though the mechanism of this interaction remains incompletely defined [14–18].

The DDK phosphorylates the MCM complex, an early step in the conversion of this complex into the two active helicases that unwind DNA. In the simplest version of this model, early origins exist in chromatin regions enriched for FKH sites thus promoting higher concentrations of Fkh proteins that in turn recruit the DDK [11,14,18]. Thus, early origins are exposed to a higher local concentration of DDK compared to late origins at the start of S-phase, and hence are more likely to fire.

However, some evidence indicates that Fkhs may also regulate the G1-phase licensing step of the origin cycle. For example, Fkhs interact with ORC and/or the MCM complex, and Fkh-origin association is enhanced in G1-phase [10,11,19]. A specific motivation for this study was the identification of a subset of yeast origins, named FHA-dependent origins, which require the forkhead associated (FHA) domain of Fkh1 for normal levels of ORC binding [12]. FHA domains are conserved protein-binding modules with a distinctive specificity for peptides that contain a phosphorylated threonine (pT) [20]. FHA domains also recognize other residues adjacent to the pT within the peptide target, but their identities are distinct for each FHA domain. Thus, pT recognition is the defining functional characteristic of FHA domains. A key conserved arginine within FHA domains is critical for pT recognition, and its substitution abolishes FHA-dependent protein-protein interactions. In yeast, *fkh1-R80A* is the relevant allele, and inactivates the Fkh1-FHA domain [21,22]. The *fkh1-R80A* mutant reduces the activity of FHA-dependent origins, while the activity of origins within a control group (referred to as FHA-independent) are unaffected or even enhanced [12]. As a group, FHA-dependent origins are characterized by a FKH site in a T-rich orientation positioned 5' of the origin's essential ORC site (5' FKH-T). Mutation of the 5' FKH-T motif within several selected FHA-dependent origins abolishes their Fkh1-FHA domain-dependent stimulation, providing genetic evidence that the Fkh1-FHA domain performs its role at these origins through a single, discrete FKH site [12]. This 5' FKH-T site requirement distinguishes FHA-dependent origins from the Fkh1/2-activated origins that have been examined to date that require FKH motifs in the opposite orientation and positioned 3' of the essential ORC site (3' FKH-A). These different motif requirements raise the possibility that Fkhs can promote origin activity through multiple mechanisms. FHA-dependent origins were defined parsimoniously by the ability to experimentally assign them a distinct ORC-origin recognition mechanism [23]. As such, this origin group constitutes <5% of yeast genomic origins. Thus, it was unclear whether the impact the Fkh1-FHA domain had on FHA-dependent origins would apply to other yeast origins.

In this report, the Fkh1-FHA domain's impact on chromosomal origin activity and ORC-origin interactions was addressed at the genome scale using S-phase SortSeq (henceforth Sort-Seq), ORC ChIPSeq and MNaseSeq experiments to compare *FKH1* and *fkh1-R80A* mutant yeast. The SortSeq data provided evidence that the Fkh1-FHA domain was required for the normal activity of approximately half of all yeast chromosomal origins, acting as a positive regulator of most origins that typically fire in early S-phase (early origins; termed FHA-SORT-positive origins), and a negative regulator of many late origins (FHA-SORT-negative origins). A significant number of FHA-SORT-positive origins were also identified previously in a BrdU ChIP-chip experiment as Fkh1/2-activated origins [10], but there were also distinct origins in

the FHA-SORT-positive group. Specifically, while Cen-associated origins were not defined as Fkh1/2-activated origins, they were identified as FHA-SORT-positive origins, suggesting that features of Cen-associated origin control remain undefined [5,17,24]. ORC ChIPSeq data provided evidence that normal levels of ORC-origin binding at FHA-positively regulated origins required the Fkh1-FHA domain. MNaseSeq experiments provided evidence that the Fkh1-FHA domain promoted the stability of nucleosomes adjacent to both origins and promoters in G1-arrested cells while reducing their stability in proliferating cells, providing evidence for a global, cell-cycle regulated role of the Fkh1-FHA domain in normal nucleosome behavior within key genomic regulatory regions. In contrast, higher-resolution analyses of nucleosome and smaller ORC-sized DNA fragments provided evidence that the Fkh1-FHA domain promoted chromatin architectural features distinct to G1-specific origin-associated chromatin. Thus, the yeast Fkh1-FHA domain controlled origin activity, normal ORC-origin interactions and G1-specific hallmarks of origin-associated chromatin at a substantial fraction of yeast chromosomal origins.

## Results

### The Fkh1-FHA domain contributed to normal replication of the yeast genome during an unperturbed cell cycle

SortSeq experiments were used to examine genome replication in proliferating yeast populations. In this approach, the yeast cells are harvested, fixed, stained and sorted into S-phase and late G2-phase populations [25] (S1 Fig). The normalized S-phase DNA copy numbers for each 1 kb genomic region were determined and used to generate chromosome replication profiles (Fig 1A). SortSeq data allows visualization of major replication intermediates, with peaks indicating origins, slopes indicating replication forks, and valleys indicating termination zones. Fig 1A shows the replication profile obtained for chromosome II in *FKH1* (black) and *fkh1-R80A* mutant (red) cells. All yeast chromosomes are shown in S2 Fig. The replication profiles obtained for *FKH1* cells recapitulated outcomes from published experiments [26,27]. Because biological or technical variables might cause variation in the outcomes of these experiments, data from biologically independent SortSeq replicates of wild-type (*FKH1*) (n = 3, black) and mutant (*fkh1-R80A*) (n = 2, red) yeast were generated and processed independently. The chromosomal scan in Fig 1A depicts the mean value for each 1 kb region across the genome of a given genotype as well as the 95% confidence interval for that mean. These values are indicated as solid lines and vertical shading, respectively. The greatest consistent variation between genetically identical independent experiments occurred over origin peaks and at termination valleys, though some chromosomal-end regions also showed substantial variation (Fig 1A, black) [26]. Nevertheless, even taking into account the variation between independent replicate SortSeq experiments, the data revealed that the *fkh1-R80A* yeast altered the replication of yeast chromosomes.

Several points were noted. First, the activity of many of the most frequently used (i.e. efficient) origins, defined here as the most highest peaks in *FKH1* cells, were reduced in *fkh1-R80A* cells. Second, euchromatic origins near the chromosomal termini, in particular those defined by less prominent peaks, showed enhanced replication efficiency in *fkh1-R80A* cells. For example, the activity of at least one origin (in the vicinity of on the right end of chromosome II) was enhanced in *fkh1-R80A* cells compared to *FKH1* cells. Third, a few replication origin zones, defined as three or more ARSs within the same contiguous chromosomal region (within $\leq$ 30 kbp [24,28]), showed peak broadening in the SortSeq data. For example, the intrinsic probability of *ARS201.7* and *ARS203* increasing relative to *ARS202* could explain the broadening of the peak in this region (Fig 1A). Fourth, the behavior of some termination

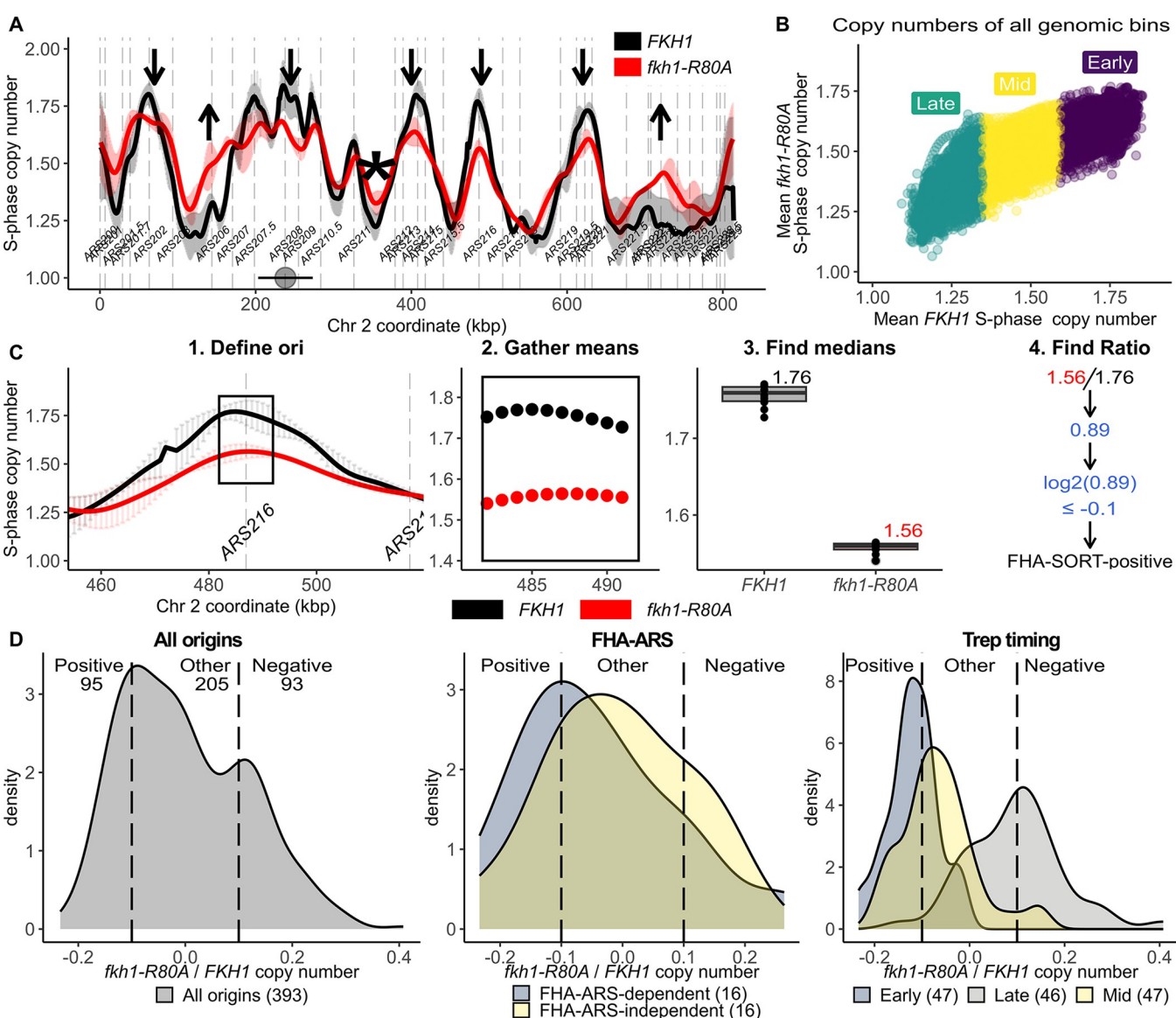

**Fig 1. The Fkh1-FHA domain contributed to normal replication of the yeast genome during an unperturbed cell cycle. (A)** Normalized S-phase copy numbers across chromosome II for two independent *fkh1-R80A* mutant cell populations (red) and three independent *FKH1* populations (black). The solid lines are the mean obtained for each 1 kb region assessed from each experiment, and the shaded vertical lines indicate the 95% confidence interval for that mean. Downward arrows mark origins whose activity is reduced in *fkh1-R80A* cells, while upward pointing arrows mark origins whose activity is enhanced. The * marks a termination zone whose replication has been altered substantially by the *fkh1-R80A* allele. **(B)** The mean value S-phase copy numbers for every 1 kb bin of the *fkh1-R80A* mutant (y-axis) are plotted against their value in FKH1 cells (x-axis) **(C)** The approach used to quantify the impact of the *fkh1-R80A* allele on each origin across the yeast genome is depicted for FHA-positive origin *ARS216*. First a 10 kb fragment centered on the T-rich start of the defined ORC site was selected (1, boxed region). Next, the mean S-phase value for each 1 kb region across this 10 kb origin fragment for *FKH1*, (black) or *fkh1-R80A* (red) cells was determined (2, each dot represents a distinct 1 kb region within the 10 kb origin locus). Next, the median value of the 10 means was determined for each origin and assigned as a distinct S-phase value in either *FKH1* or *fkh1-R80A* cells (3). Finally, the log2 of each origin's *fkh1-R80A*/*FKH*1 S-phase copy number ratio was determined (4) and used in subsequent graphs. **(D)** The distribution log2(*fkh1-R80A*/*FKH1*) values for each origin was summarized by smoothed kernel density estimates (KDE) for the indicated origin groups. KDE plots of origin number (density, y-axis) versus log2(*fkh1-R80A*/*FKH1*) ratios (x-axis) are displayed. 'All origins' refers to the 393 confirmed origins from the 410 origins defined in [48] for which we could assign a high-confidence ORC site [12]. 'FHA-ARS' refers to the 32 origins characterized in a previous study as described in [12]. 'Trep timing' refers to the 238 origins assigned a replication time (Trep) as measured for a synchronous S-phase as in [29]. Origins with log2(*fkh1-R80A*/*FKH1*) values ≤ -0.1 were considered positively regulated by the Fkh1-FHA domain (FHA-SORT-positive), while those with log2(*fkh1-R80A*/*FKH1*) values ≥ +0.1 were considered negatively regulated (FHA-SORT-negative). These cut-offs are indicated on the KDE plots with vertical lines. All origins that failed to meet these cut-offs were placed in the 'Other' category. **(D)** Quantitative analysis of the effect of the *fkh1-R80A* allele on duplication of the entire yeast genome. The genome was parsed into 11,800 1 kb regions. The mean S-phase copy number for each region in *fkh1-R80A* cells (y-axis) was plotted against the mean of its S-phase copy number in *FKH1* cells (x-axis). The colors indicate the regions that replicate in the first (early, purple), middle (mid, yellow) or last (late, green) third of S-phase.

zones was altered, an expected byproduct of altered origin use within the population [26]. For example, the replication of the termination zone between *ARS202* and *ARS207.5*, which is one of the last regions of chromosome II to be duplicated in *FKH1* cells, was enhanced in *fkh1-R80A* cells, presumably as a consequence of the enhanced activation probability of two inefficient origins, *ARS206* and *ARS207.5*.

Substantial alterations in the distribution of origin activity across chromosomes should alter the normal spatiotemporal pattern of genome duplication. To generate a quantitative test of this expectation, the replication of the entire yeast genome in *fkh1-R80A* mutant cells was plotted against its replication in *FKH1* cells (Fig 1B). Specifically, the genome was divided into 11,800 1 kb regions, and the S-phase copy number for each region in *fkh1-R80A* cells (y-axis) was plotted against its S-phase copy number in *FKH1* cells (x-axis). This analysis revealed that the *fkh1-R80A* allele mitigated the differential between the yeast genome's earliest and latest replicating regions, flattening the temporal landscape observed in *FKH1* cells.

## The Fkh1-FHA domain promoted the activity of a substantial fraction of early and Fkh1/2-activated origins

For a more quantitative assessment of the impact of the *fkh-1R80A* allele on yeast chromosomal origin activity, each confirmed origin was assigned a log2(*fkh1-R80A/FKH1*) S-phase copy number ratio (Fig 1C). These values were then summarized in Kernel Density Estimation (KDE) graphs of origin density (y-axis) versus the log2(*fkh1-R80A/FKH1*) S-phase copy number ratio (x-axis) for all confirmed origins (Fig 1D). This graph revealed that the activity of many yeast origins was altered by the *fkh1-R80A* allele. For subsequent analyses, the origins were divided into three distinct groups: 1. FHA-SORT-positive origins, which showed 10% or more reduced activity in *fkh1-R80A* cells; 2. FHA-SORT-negative origins, which showed 10% or more enhanced activity; 3. FHA-SORT-other, comprising all origins not meeting either of these cut-offs (Fig 1D, left panel). Next, KDE analysis was applied to the FHA-ARS-dependent and FHA-ARS-independent origin subsets defined in our previous studies [12,23] (Fig 1D, middle panel). Each subset consists of 16 distinct origins that were initially defined by their ORC-origin binding mechanisms and subsequently parsed by the effect of the *fkh1-R80A* allele on their ARS activity. Specifically, FHA-ARS-dependent origins are defined as those origins among the 32 assessed whose ARS activity (i.e. ability of the origin to support plasmid replication) is reduced in *fkh1-R80A* cells. Conversely, FHA-ARS-independent origins are those whose ARS activity is unaffected (or even enhanced) in *fkh1-R80A* cells. KDE analyses revealed that the FHA-ARS-dependent origin group was shifted left, in the direction of FHA-positive regulation, whereas the FHA-ARS-independent origin group was shifted right, in the direction of FHA-SORT-negative regulation, an expected outcome based on the effect of the *fkh1-R80A* allele on the ARS activity of these origins [12]. However, considerable overlap between these two groups was observed. Therefore an origin-by-origin examination of the SortSeq data was applied to FHA-ARS-dependent and FHA-ARS-independent origins (S3 Fig). This analysis provided at least a partial explanation for the substantial overlap observed for these two origin groups' behaviors in their chromosomal contexts (Fig 1D). In particular, these origins were initially classified based on their ORC-origin binding properties, not their chromosomal origin activity [23]. Some of the FHA-regulated ARSs were relatively inactive in their chromosomal contexts likely due to their proximity to neighboring origins. Thus, there were limitations to using the log2(*fkh1-R80A/FKH1*) metric for assigning individual origins discrete activity values. Nevertheless, despite this limitation, the origin-by-origin assessment in S3 Fig revealed that the majority ($\approx$ 65%) of FHA-ARS-dependent origins, but only a minority ($\approx$ 33%) of FHA-ARS-independent origins, showed reduced origin activity on the chromosome.

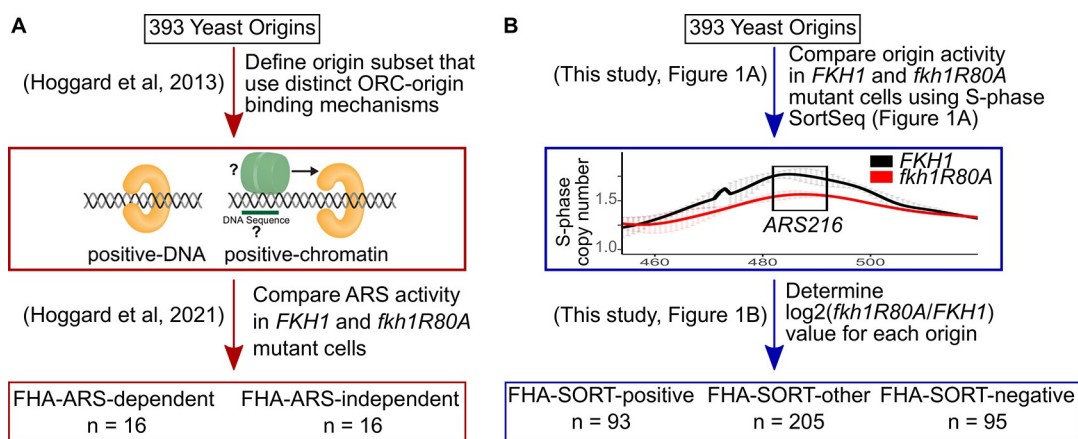

**Fig 2. Summary of the functionally defined Fkh1-FHA-regulated origin groups relevant to this study. (A)** The Fkh1-FHA-ARS regulated origins were defined over the course of previous studies [12,23,30]. First, a collection of two origin groups that used distinct mechanisms for ORC-origin binding were defined. At positive-DNA origins the *in vivo* ORC-origin interaction can be explained by the strength of the intrinsic ORC-origin DNA binding interaction *in vitro*. In contrast, positive-chromatin origins display weak ORC-origin DNA binding interactions *in vitro*. Therefore, a model for ORC binding to positive-chromatin origins is that they are associated with a feature(s) extrinsic to the ORC site that promote ORC binding *in vivo*. In the simplest direct scenario, an origin-adjacent DNA sequence binds a factor that interacts with ORC to aid in its binding to origin DNA. These origin collections were then assessed using an ARS assay in *fkh1-R80A* and *FKH1* cells. Out of 32 origins assessed, 16 showed reduced activity in *fkh1-R80A* cells (FHA-ARS-dependent) and 16 were relatively unaffected or showed enhanced activity (FHA-ARS-independent). An origin producing a ≥2-fold reduction in ARS activity in *fkh1-R80A* cells is defined as an FHA-ARS-dependent origin. The majority (75%) of FHA-ARS-dependent origins are also positive-chromatin origins, while the majority (75%) of FHA-ARS-independent origins are positive-DNA origins. This division was potentially a slight underestimate of the linkage between ORC-origin binding mechanism and FHA-dependent ARS activity. Specifically, *ARS516* is assigned to the FHA-ARS-independent group, while based on the analyses in S3 Fig, it was better aligned with the FHA-ARS-dependent group. Thus, 76.5% of FHA-ARS-dependent origins are also positive-chromatin and 73.3% of FHA-ARS-independent origins are positive-DNA origins. Regardless, FHA-ARS-activity was linked to ORC-origin binding mechanisms within these origin subgroups. **(B)** FHA-SORT-regulated origins were defined solely based on the effect the *fkh1-R80A* allele had on their normalized S-phase copy number, as described in Fig 1.

Visual inspection of chromosome II (Fig 1A) and the role of the Fkh1-FHA domain in promoting the differential replication of early and late replicating regions (Fig 1B) indicated a link between the Fkh1-FHA domain and an origin's replication time. Therefore KDE analysis was applied to origin groups parsed by their experimentally measured S-phase replication time (Trep values) [29]. This analysis revealed that the activity of most early origins was reduced in *fkh1-R80A* cells, while the activity of many late origins was enhanced (Fig 1D, right).

For downstream analyses, two types of origin classifications were considered further (Fig 2). First, two small subsets of experimentally examined origins classified previously based on the role of the Fkh1-FHA domain on their ARS activity were considered (Fig 2A, FHA-ARS-defined) [12]. Second, the substantially larger origin subsets classified based on their activity as measured in SortSeq experiments, as described above, were also considered (Fig 2B, FHA-SORT-defined). Thus, more than half of early origins and a substantial fraction of origins that act in mid S-phase were referred to as FHA-SORT-positive (n = 93) to distinguish them from the previously examined positively regulated subset of FHA-ARS-dependent origins (n = 16). In contrast, origins whose activity was enhanced in *fkh1-R80A* cells as measured in SortSeq experiments were named FHA-SORT-negative origins (n = 95) to distinguish them from the previously examined negatively regulated FHA-ARS-independent origins (n = 16). About half of late origins were FHA-SORT-negative.

The strong link between Fkh1-FHA origin regulation and origin replication time provided evidence that Fkh1-FHA regulation revealed by SortSeq explained, at least to some extent, Fkh1/2 regulation originally defined in [10]. The degree of overlap between the FHA-SORT-

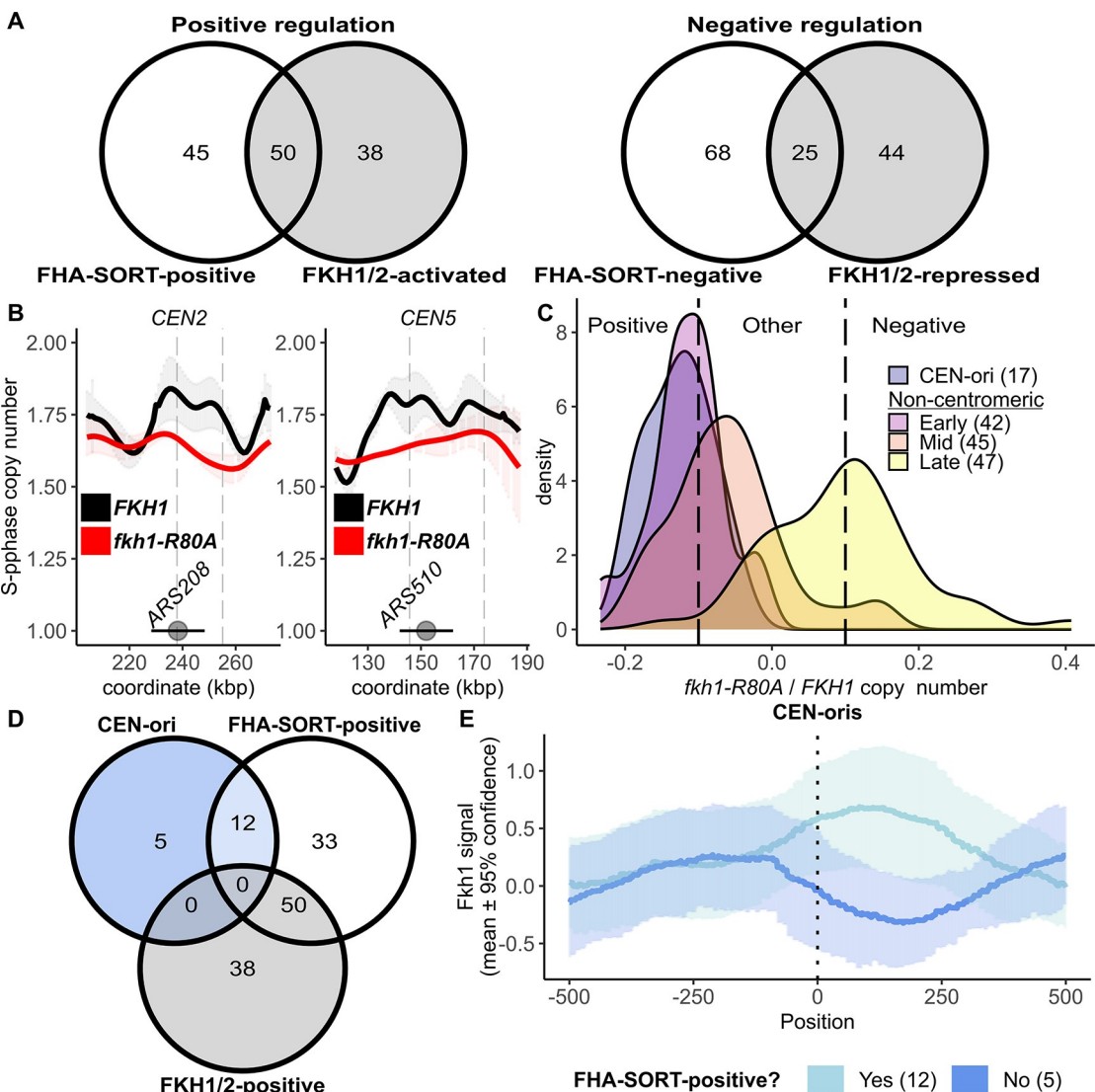

**Fig 3. The FHA-SORT-positive and Fkh1/2-activated origin groups overlapped significantly, but the FHA-SORT-positive group included centromere-associated origins. (A)** Overlap between the FHA-SORT-positive origins and the FHA-SORT-negative origins. The Fkh1/2-activated and Fkh1/2-repressed origins were defined previously by BrdU incorporation followed by IP [10]. **(B)** SortSeq scans over two centromeres that had FHA-positive origins. For scans of all centromeres see S4 Fig. The centromeres are indicated by gray circles, and the horizontal line indicates the 10 kbp region that was used to identify Cen-associated origins. The positions of confirmed origins are indicated with gray, dashed vertical lines. **(C)** The KDE graph includes 17 Cen-associated origins as a separate category. **(D)** The overlap among the 17 centromere-associated origins as defined in this study, and FHA-SORT-positive and Fkh1/2-activated origins. **(E)** An Fkh1 ChIP-chip dataset published in [12] was used to assess Fkh1-ChIP signals at Cen-associated origins that were either called as FHA-SORT-positive (light blue, n = 12) or not (blue, n = 5). The IP/input ratio for each nucleotide across the 10 kb span was determined, and the mean IP/input values for all nucleotides across all origins in the group are shown as a line, while the 95% confidence interval for each mean value is indicated by shading, as described previously [12].

regulated origin groups defined in this study and the Fkh1/2-regulated origin groups defined previously was addressed in Fig 3A. A substantial fraction, 53%, of the FHA-SORT-positive origins were previously identified as Fkh1/2-activated origins. In contrast, minimal overlap was observed between the FHA-SORT-negative and Fkh1/2-repressed origin groups. Statistically, the two different negatively regulated groups were unrelated. Regardless, the overlap between FHA-SORT-positive and Fkh1/2-activated origins was significant, particularly

considering the substantial differences in the experimental approaches used to define them. Thus, the Fkh1-FHA domain accounted for much of the origin specificity assigned to Fkh1/2 positive regulation.

## The Fkh1-FHA domain promoted the activity of centromere-associated origins

Cen-associated origins were a specific example of a difference between this study's FHA-SORT-positive origins and the Fkh1/2-activated origin collection defined previously [10]. Cen-associated origins fire in early S-phase due to recruitment of the DDK by the Ctf19 kinetochore complex [17,24], and are a small subset of early origins that are not Fkh1/2-activated [5]. In the SortSeq experiments, however, many Cen-associated origins qualified as FHA-SORT-positive (Figs 3B and 3C, and S4 Fig). Indeed, based on KDE analyses, Cen-associated origins were among the the most FHA-SORT-positive origins (Fig 3C). 17 Cen-associated origins were defined as FHA-positive based on a stringent criterion (origins within a 10 kb span 5' or 3' of the defined centromere) [24]. Twelve of these were FHA-SORT-positive, while none were called as Fkh1/2-activated (Fig 3D).

   Current models posit that Fkh proteins promote origin activity by binding near these origins' ORC sites. Therefore, Fkh1 binding near Cen-associated origins was examined using genomic Fkh1 ChIP datasets (Fig 3E). Fkh1 ChIP signals at FHA-SORT-positive, Cen-associated origins were detected over DNA regions typically required for full origin activity, demarcated by the ORC start site (T-rich strand) and DNA sequence 3' of that site. In contrast, Fkh1 ChIP signals were relatively depleted at the Cen-associated origins that were not affected by *fkh1-R80A*.

## FHA-SORT-defined and FHA-ARS-defined origins had distinct FKH motif organizations

The SortSeq experiments presented here revealed that $\approx 25\%$ of yeast origins were FHA-SORT-positive. Thus, the FHA-ARS-dependent origins defined previously represented only a small fraction of yeast origins that were sensitive to the Fkh1-FHA domain. FHA-ARS-dependent origins were notable for their association with a FKH motif match positioned proximal to and 5' of their ORC sites, and in the same T-rich orientation as their ORC sites (5' FKH-T). The requirement for this 5' FKH-T in Fkh1-FHA-dependent ARS activity is supported by experimental data [12]. In contrast, independent studies that focus on selected Fkh1/2-activated origins provide evidence that these origins rely on 3' FKH-A motifs [10,19]. To explore FKH site organization at the distinct collections of origins considered in this report (see Fig 2), a stringent, experimentally-aided definition for a FKH1 motif was used to assess both the orientation and positioning of FKH1-motifs associated with relevant origin groups (Fig 4).

   There were over 130,000 matches to the 8 bp AT-rich FKH1 motif derived from analyses of the Fkh1 ChIP-chip binding sites that reached a P-value threshold of $10^{-3}$ within the yeast genome [12]. To establish a more stringent collection of FKH motifs, only the top 25% of FKH1 motifs that exceeded this $10^{-3}$ P-value threshold were considered, reducing the number of matches to 34,066. Thus defined, the FHA-SORT-positive and FHA-SORT-negative origins were assessed for FKH1-T and FKH1-A matches over a 501 bp span centered over the start position, "0", of the T-rich strand of the ORC site (boxed) (Fig 4A). Visual inspection revealed that FHA-SORT-positive origins contained a higher density of FKH1 motifs compared to FHA-SORT-negative origins, as was noted for Fkh1/2-activated versus Fkh1/2-repressed origins [10]. However, in contrast to the FHA-ARS-dependent origins defined previously [12], an enrichment for 5' FKH1-T motifs at FHA-SORT-positive origins compared to FHA-SORT-

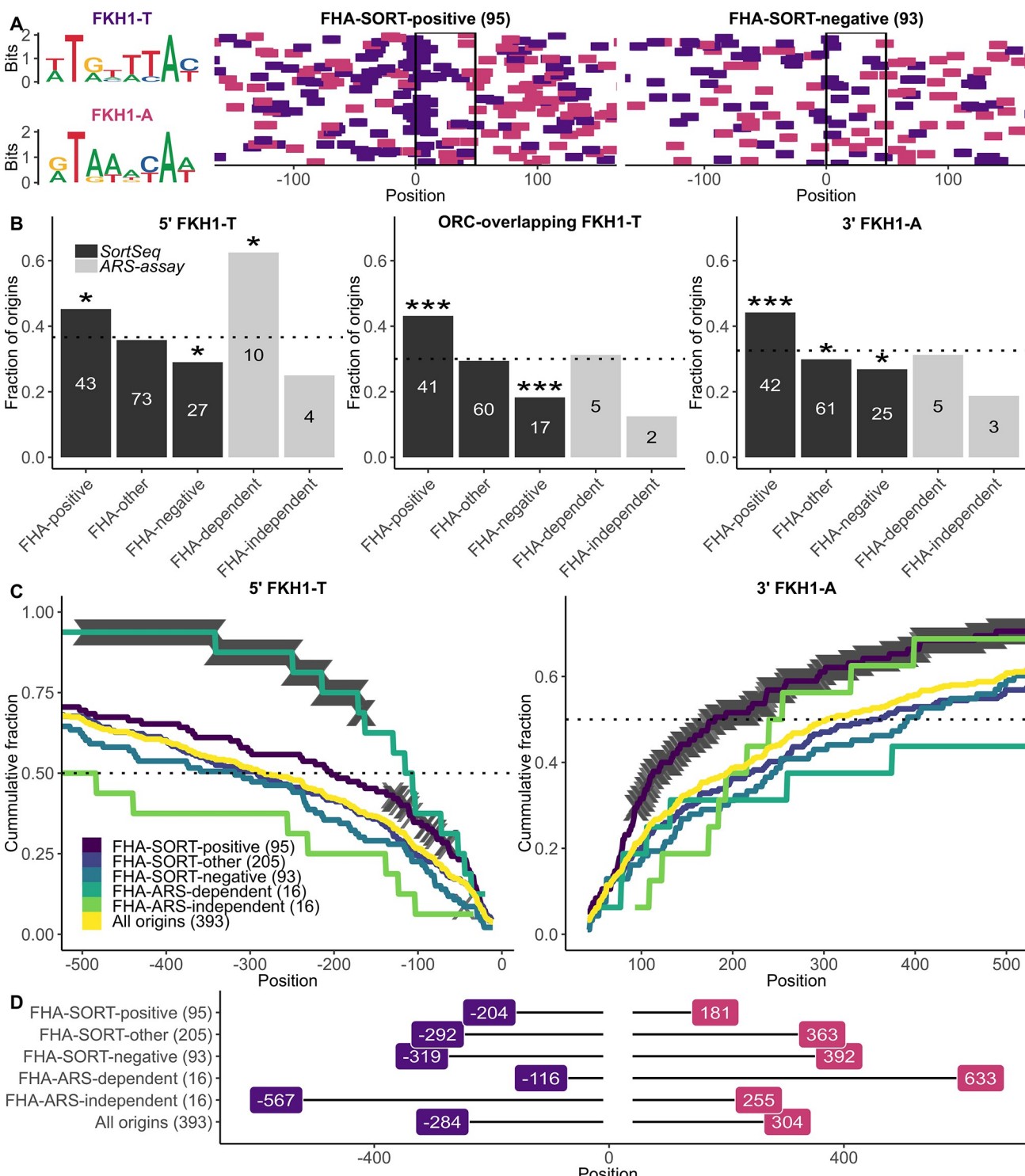

**Fig 4. FHA-SORT-defined and FHA-ARS-defined origins were associated with distinct FKH motif organizations. (A)** Sequence logos of the FKH motifs in both orientations were mapped at FHA-SORT-positive and FHA-SORT-negative origins (Fig 1) as purple (FKH1-T) or pink dashes (FKH1-A). The origins in each group are aligned with respect to their ORC sites (T-rich strand), the start (nucleotide 0) and final (nucleotide +32), indicated by the black box. **(B)** The fraction (y-axis) of origins within the indicated group (x-axis) containing at least one match to the indicated motif within the indicated origin regions for the relevant FHA-regulated origin groups classified by SortSeq in this study or by ARS assays previously [12] (see Fig 2). "5'FKH1-T" queried nucleotides -150 through -11 for FKH1-T motif matches, while "3'FKH1-A" queried nucleotides +41 through +150 for FKH1-A matches. The "ORC-overlapping" region encompassed nucleotides -10 through +40 and was queried for FKH1-T motifs. The black bars refer to origin groups classified by Sortseq in this study (Fig 2B). Gray bars refer to the smaller subsets of origins classified previously (Fig 2A) [12]. The horizontal line

indicates the fraction of all confirmed origins (n = 393) that contained a match to the queried FKH1 motif in the origin region under assessment. The enrichment or depletion of a given motif in any given origin group was challenged against the fraction of that motif in all confirmed origins using the hypergeometric distribution function. Significant P-values are denoted by asterisks (*, P < 0.05; **, P < 0.001; ***, P < 0.0001). In these analyses, 150 bp regions 5' and 3' of the ORC site were queried. In the previous study [12], 250 bp were queried for these regions. This difference explained why only 10 FHA-ARS-dependent origins contained a 5'FKH-T in this study versus 12 in the previous study. **(C)** The cumulative fraction of origins (y-axis) in the indicated groups containing a 5'FKH-T (left) or a 3'FKH-A (right) after traversing the indicated number of nucleotides from the ORC site (x-axis). Thus 50% of all FHA-ARS-dependent origins contained an FKH-T site within the first 116 nucleotides 5' of the ORC site, while 50% of all confirmed origins did not reach this level of FKH-T site accumulation until 284 base pairs. Nucleotide positions that reached P-value significance values of ≤ 0.01 are indicated by gray cross marks derived from hypergeometric distributions where at each position, the fraction of origins in the queried collection that contained a match by that nucleotide position was reached is compared to the fraction of all confirmed origins (n = 393) that contained a match by the same position. **(D)** Summary of the 50% accumulation point for the analyses in (C).

negative origins was not visually obvious. To assess the frequency of the two orientations of FKH1 matches adjacent to selected origin groups, the fraction of origins within each of the indicated groups that contained the queried FKH1 motif was determined for each of the following three origin regions (Fig 4B): 1. FKH-T motifs positioned 5' of the ORC site (comprising nucleotides -150 through -11); 2. FKH1-A motifs positioned 3' of the ORC site (nucleotides +41 through +150) and 3. FKH1-T motifs overlapping the ORC site (nucleotides -11 through +41). Fig 4B displays only the origin-region/FKH1 motif combinations that generated statistically significant origin-group enrichments or depletions relative to all confirmed origins (n = 393; Fig 4B, indicated with dotted line in these panels). The outcomes for the smaller groups of FHA-ARS-dependent and FHA-ARS-independent origins were reported previously but were also included here for comparison (gray bars) [12].

As reported previously, FHA-ARS-dependent origins were substantially enriched for a 5' FKH1-T match compared to both FHA-ARS-independent and all confirmed origins [12] (Fig 4B and 5' FKH1-T). FHA-SORT-positive origins also showed a modest enrichment for a 5' FKH1-T motif relative to all confirmed origins, while FHA-SORT-negative origins showed a modest depletion of this motif. However, the differential presence of a 5' FKH1-T motif that was so striking for the FHA-ARS-dependent versus FHA-ARS-independent group comparison was modest for the FHA-SORT-positive versus FHA-SORT-negative origin group comparison. Other differences between the SortSeq- and ARS-defined FHA-regulated origins were noted. In particular, while FHA-ARS-dependent origins showed no enrichment for 3' FKH1-A sites compared to confirmed origins, the FHA-SORT-positive origins were substantially enriched for this motif (Fig 4B and 3' FKH1-A).

The largest differential presence of an FKH motif between FHA-SORT-positive and FHA-SORT-negative origins was a FKH1-T match overlapping the ORC site (Fig 4B, ORC-overlapping FKH1-T). Such an overlap is also enriched in the collection of Fkh1/2-activated origins [10]. Specifically, FHA-SORT-positive origins were 2.4x as likely to have an ORC-site-overlapping FKH1-T motif compared to FHA-SORT-negative origins. While the numbers were too small to generate P-value significance values for the FHA-ARS-dependent and FHA-ARS-independent group comparisons, the former group was 2.5 times as likely as the latter group to contain an ORC site-overlapping FKH1-T motif. Thus, while the differential presence of a 5' FKH1-T motif was distinct to the FHA-ARS-dependent/-independent origin group comparison, the differential presence of an ORC-site-overlapping FKH1-T site was shared by both the FHA-SORT-positive/-negative and the FHA-ARS-dependent/-independent origin group comparisons.

While the analyses in Fig 4B revealed differences in FKH1 motif organization associated with these different origin groups, even the most substantial effects were due to the behavior of fewer than half of all origins within the queried category. Therefore, motif accumulation analyses were performed, wherein the cumulative fraction of origins (y-axis) that contained the

queried motif between the indicated nucleotides (x-axis) for 500 bp regions 5' and 3' of the origin ORC sites was determined (Fig 4C). The nucleotide position where 50% of all origin fragments within the queried origin group had acquired a match to the indicated FKH1 motif is summarized in Fig 4D. The outcomes from these analyses solidified and extended the conclusions of Fig 4B. Specifically, a robust enrichment for a proximal 5' FKH1-T as well as a contrasting depletion of this motif was a distinctive characteristic of the FHA-ARS-dependent/-independent origin group comparisons [12]. In contrast, for SortSeq classified origins, a robust enrichment of a downstream, proximal FKH1-A motif was the more differentiating characteristic of the FHA-SORT-positive/-negative origin group comparison. This outcome provided evidence that the FHA-ARS-regulated group was enriched for a particular Fkh1-FHA-regulated mechanism (i.e. dependent on a 5' FKH-T), while more than a single mechanism was likely operating at FHA-SORT-regulated origins. Because many Cen-associated origins were identified as FHA-SORT-positive, FKH1 motif analyses were also performed for these origins as a separate group (S5 Fig). While the origin numbers were small, these analyses provided evidence that FHA-SORT positive Cen-associated origins, at least in terms of FKH1 motifs, differed from the majority of FHA-SORT-positive origins.

## The FKH1-FHA domain promoted ORC association at positively regulated origin groups

The Fkh1-FHA domain promotes ORC binding at the subset of positive chromatin origins that are FHA-ARS-dependent [12] (see Fig 2). Of course, these origins represent only a tiny fraction of yeast origins. In addition, they were intentionally selected for further study because they displayed an experimentally definable, and thus potentially extreme, ORC-origin binding mechanism *in vivo* [23,30]. However, the SortSeq data presented above provided evidence that many yeast origins outside of this small collection were dependent on the Fkh1-FHA domain for their normal activity. To assess how the Fkh1-FHA domain influenced ORC-origin binding beyond the FHA-ARS-dependent origins, the ORC ChIPSeq signals (henceforth ORC signals) from this data set were examined at several relevant origin groups (Fig 5). Specifically, the mean ORC signal for each nucleotide position for each of the origins within the indicated groups in *FKH1* and *fkh1-R80A* samples was determined (y-axis) and plotted relative to origin position. The variation of ORC signals among the individual origins within each of the positively regulated groups was substantial, the more robust effect of the *fkh1-R80A* allele on positively regulated origins compared to the negatively regulated origins was clear (S6 Fig).

A comparison of the complete collection of FHA-ARS-dependent and FHA-ARS-independent origin groups revealed that the FHA-ARS-dependent group (i.e. positively regulated) generated a higher mean ORC signal compared to the FHA-ARS-independent group (i.e. negatively regulated) in *FKH1* cells, consistent with previous conclusions (Fig 5A and 5B). However, the *fkh1-R80A* mutation caused a greater reduction in the ORC signal at the FHA-ARS-dependent group than at the FHA-ARS-independent group, consistent with previous independent analyses [12]. These data could only be interpreted in terms of relative effects on ORC binding at these two origin groups. Nevertheless, they provided evidence that the *fkh1-R80A* allele altered the distribution of ORC between these two subsets of origins. To further illustrate this point, in Fig 5B, ORC signals at the two contrasting origin groups for *FKH1* or *fkh1-R80A* are depicted in the same graphs. In *FKH1* cells the ORC signals at the FHA-ARS-dependent group are greater than they are at the FHA-ARS-independent group. In contrast, in *fkh1-R80A* cells, the ORC signals at these two distinct groups were similar [12,23].

These analyses were applied to the ORC signals generated by the larger groups of FHA-SORT-regulated origins (FHA-SORT-defined, Fig 5C and 5D) as well as the Fkh1/2-regulated

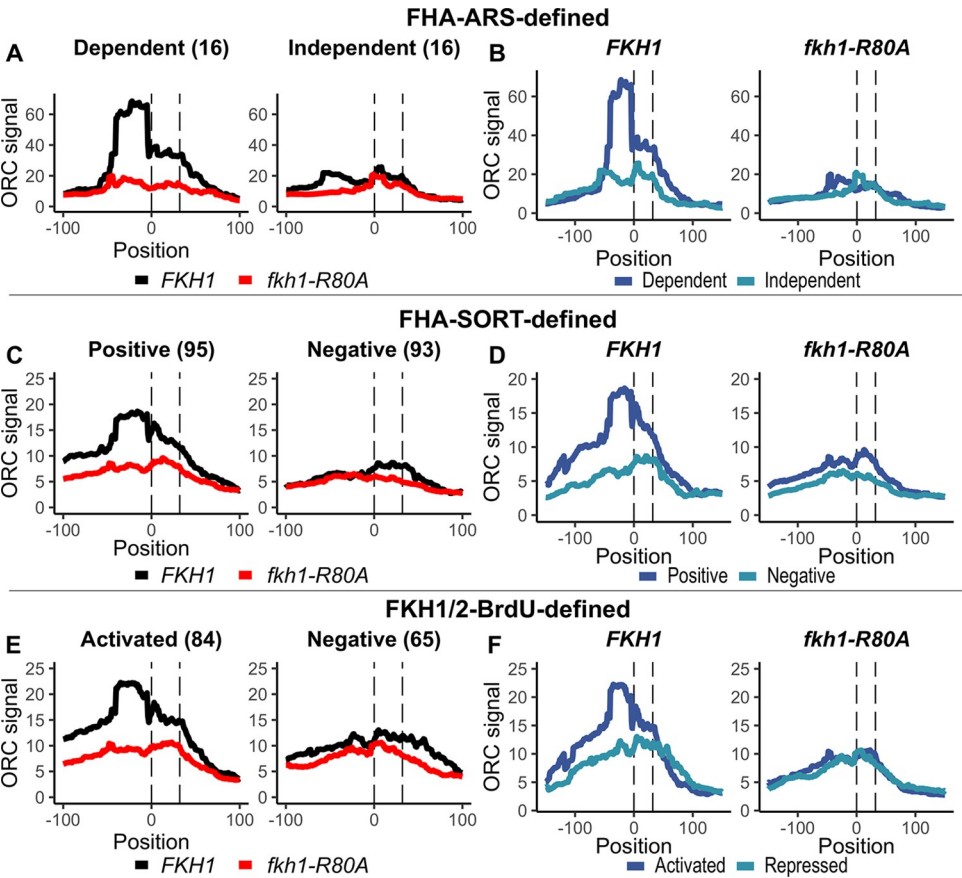

**Fig 5. The Fkh1-FHA domain promoted normal ORC ChIPSeq signals at positively-regulated origin groups.** ORC ChIPSeq data [12] was assessed at the indicated origin groups by plotting the per-nucleotide mean signal for the internally normalized ORC ChIPSeq signals (95% confidence intervals are included in S5 Fig) across the spans of the indicated origin groups, aligned based on the T-rich strand of their ORC sites, with '0' marking the first nucleotide of the site. **(A)** FHA-ARS-dependent and -independent origins in *FKH1* (black) and *fkh1-R80A* (red) cells (see Fig 2A). **(B)** The same data as in panel (A) except with both FHA-ARS-dependent (dark blue) and independent (light blue) signals plotted on the same graph for either *FKH1* or *fkh1-R80A* cells. Panels **(C)** and **(D)** as in (A) and (B), respectively, except for FHA-SORT-defined origins (see Fig 2B). Panels **(E)** and **(F)** as in (A) and (B), respectively, except for FKH1/2-BrdU-defined origins as in [10].

origins defined in [10] (Fkh1/2-BrdU-defined, Fig 5E and 5F). While both of these origin groups were defined by measures of origin activity, not ORC binding, their analyses generated similar conclusions. Specifically, the *fkh1-R80A* allele caused a more substantial reduction in ORC signals at the positively regulated groups (FHA-SORT-positive in Fig 5C and 5D or BrdU-activated in Fig 5E and 5F) compared to the contrasting negatively regulated groups.

These ORC ChIPSeq datasets used here were generated from crosslinked genomic DNA that had been treated by both sonication and MNase digestion to generate small fragments [31]. Perhaps the analyses of small fragments contributed to a qualitative difference also noted in these data. In particular, for the positively regulated origins in any category considered, the maximal apex of the ORC signal occurred 5' of the ORC site in *FKH1* cells. This 5' shift was either not evident or attenuated for the negatively regulated origins, albeit their overall ORC signals were low. In *fkh1-R80A* cells, the ORC signals at each of the three positively regulated collections were substantially reduced in this 5' region. While the signals directly over the ORC site were also reduced, the effect was less dramatic. An interpretation of these data is that

the *fkh1-R80A* allele led to alterations in origin architecture that in turn influenced the ORC-DNA interactions that could be trapped by the crosslinking step used in ChIP experiments.

## The Fkh1-FHA domain altered origin- and promoter-adjacent nucleosome behavior

Yeast origins generate a distinct chromatin organization at the level of nucleosome positioning, with relatively ordered nucleosomes flanking a nucleosome depleted region (NDR) centered over the ORC site [32]. This organization is disrupted by defects in ORC or by loss of several specific ATP-dependent chromatin remodelers [32,33]. MNaseSeq footprinting has been used to detect these chromatin hallmarks and to identify factors that affect them [33,34]. Here, MNaseSeq was used to examine the role of the Fkh1-FHA domain in origin-chromatin architecture. Specifically, the same strains assessed by SortSeq in Fig 2 were assessed by MNaseSeq in Fig 6, under both G1-arrested and proliferating conditions. The analysis approach taken for these MNaseSeq data is outlined in Fig 6A [35].

During the initial analyses of these data, an effect of the Fkh1-FHA domain on general nucleosome stability was noted. MNaseSeq data for all 393 confirmed origins was compared between proliferating and G1-arrested cells. Two different experiments that varied by the amount of MNase used in the digestion reactions were performed, and two independent *fkh1-R80A* mutant strains were assessed (Fig 6B and 6C). Neither origin-adjacent nucleosome placement nor NDR placement were substantially altered by the *fkh1-R80A* allele at this level of resolution and considering all 393 origins. However, in the G1-arrest experiments, the *fkh1-R80A* mutant cells generated a relative reduction in the ratio of nucleosome to subnucleosome signals compared to what was observed in *FKH1* cells. In contrast, in proliferating cells, the opposite effect was observed. Thus, normal origin-adjacent nucleosome stability during these MNase protection experiments was dependent on a functional Fkh1-FHA domain. Next, these analyses were applied to the same functional groups of origins discussed above and outlined in Fig 2 (S7 and S8 Figs). While FHA-ARS-dependent/-independent origin groups and the FHA-SORT-positive/-negative origins were functionally distinct, albeit based on different criteria (see Fig 2), each of these groups behaved similarly in these analyses. Thus, the Fkh1-FHA-dependent alterations in nucleosome stability were not linked to Fkh1-FHA-dependent origin regulation.

Promoters also generate distinct nucleosome organizations adjacent to an NDR that is positioned 5' of the transcription start site [36]. To examine the generality of the Fkh1-FHA-dependent nucleosome stability observed at origins, the MNAse-generated footprints associated with promoters were assessed (Fig 6D and 6E). These data provided evidence that the Fkh1-FHA domain also altered nucleosome stability and in similar directions at gene promoters.

## The Fkh1-FHA domain promoted a broad NDR at FHA-ARS-dependent origins in G1-arrested cells

Positioned nucleosomes framing a nucleosome depleted region (NDR) that is coincident with the ORC site and dependent on ORC-origin binding are key protein-DNA architectural hallmarks of origins [32,33,37,38]. In addition, at the more efficient origins, the ORC-defining footprint that coincides with the NDR exhibits a Cdc6-dependent extension in G1-phase [34,39,40]. To examine whether the Fkh1-FHA domain regulated these aspects of origin structure *in vivo*, the MNaseSeq data were assessed at higher resolution. For each nucleotide, the summed frequency for the relevant fragment (nucleosome-consistent (147 ± 10 bp) or ORC/

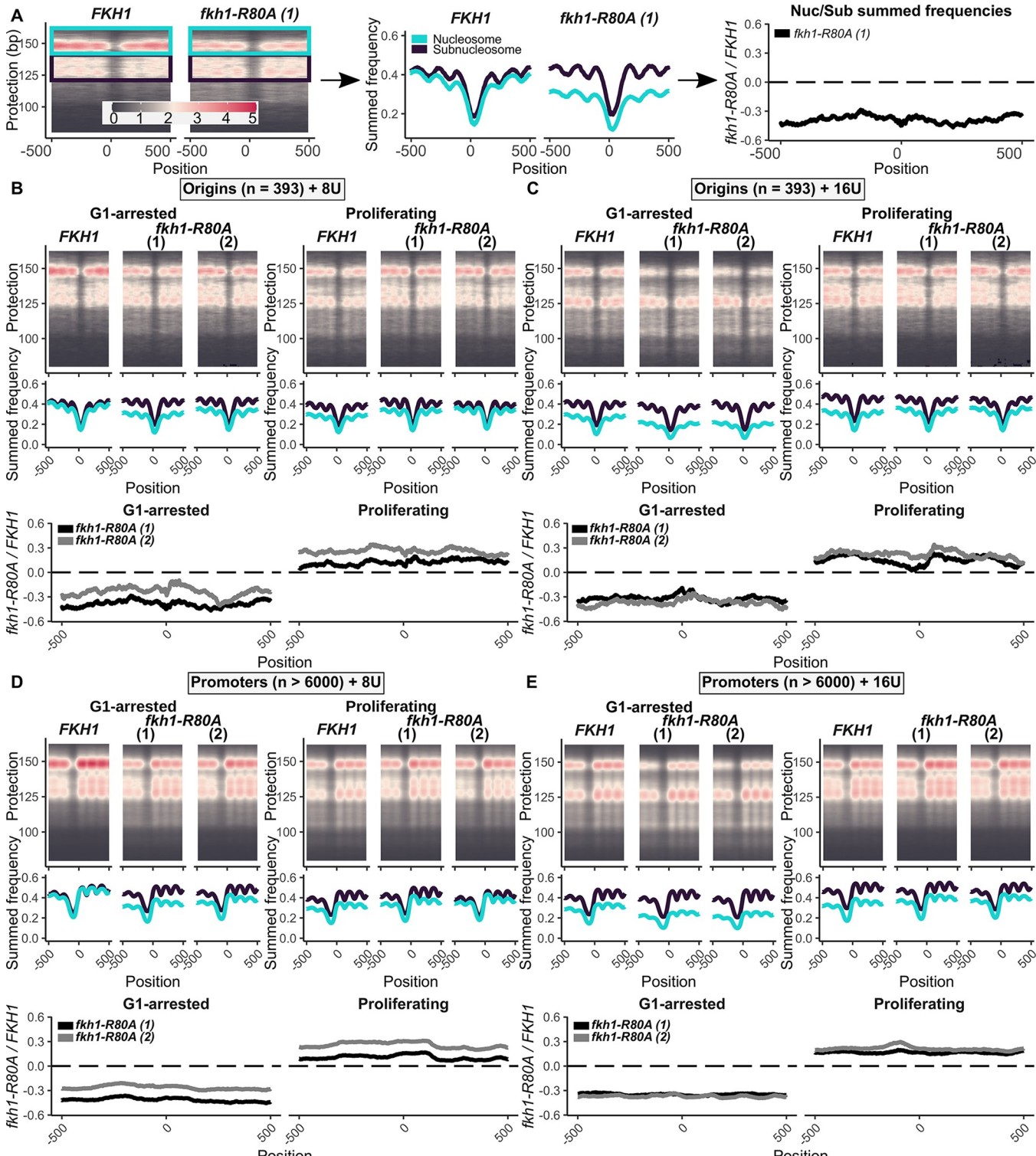

**Fig 6. The Fkh1-FHA domain altered origin- and promoter-adjacent nucleosome behavior. (A)** Plot2DO-generated heat map for the MNaseSeq data for the 393 confirmed origins [36]. The fragments consistent with nucleosome protection (142–162 bp fragments, teal) and those consistent with subnucleosome protection (121–141 bp, designated, black) are indicated. To convert these signals into graphs, the summed signals for each nucleotide on the x-axis were determined within the indicated fragment-size ranges and plotted on the y-axis for *FKH1* and *fkh1-R80A* experiments (middle panel). To quantify nucleosome stability, the per-nucleotide nucleosome:subnucleosome ratio for *fkh1-R80A* samples was divided by the same ratio determined for *FKH1* samples. **(B)** Plot2DO-generated heat maps and summed frequency graphs generated for the MNaseSeq data for the 393 confirmed origins in the indicated cell types under

G1-arrested and Proliferating conditions at the 8U MNase experiment **(C)** As in (B) for the 16U MNase experiment. **(D)** As in (B) except for gene promoters. **(E)** As in (C) except for gene promoters.

Cdc6-consistent (80 ± 10 bp)) was divided by the total signal at that same position. Each nucleotide value was scaled between 0–1 and then plotted against its origin position (S9 Fig).

A concern for these analyses was that the effect on chromatin sensitivity to MNase digestion caused by the *fkh1-R80A* mutant might make these high-resolution data too noisy to interpret. To address this concern, nucleosome signals were first compared between all origins (n = 393) and all promoters (n = 6000) under G1-arrested and proliferating conditions in both the *FKH1* and *fkh1-R80A* experiments (S9A Fig). The nucleosome signals over these loci recapitulated expectations based on published data, with a clear NDR appearing over the ORC site at origins, but 5' of the transcription start site at promoters. Importantly, the *fkh1-R80A* mutant had only minor effects on these signals under either G1-arrested or proliferating conditions. Thus, despite the substantial effects on chromatin stability noted in Fig 6, the differences between *FKH1* and *fkh1-R80A* experimental samples on nucleosome positioning were minor, suggesting that substantial differences between functionally classified origin groups could be interpreted.

A concern with analyzing the smaller ORC-consistent fragments was their low abundance. While weak signals had the potential to reflect changes in protein-DNA architecture that could be masked by the abundant nucleosome signals, they also might be challenging to decipher from noise. However, two observations of the scaled 80 ± 10 bp values in G1-arrested and proliferating cells at all origins versus all promoters suggested that these analyses reflected features of ORC-DNA architecture at origins (S9B Fig). First, only the origin signal showed a distinctive 5' shoulder, and this shoulder was present only under G1-arrested conditions. Notably, this 5' shoulder was lost in the *fkh1-R80A* mutant. Thus, the Fkh1-FHA domain was altering the G1-specific 80 ± 10 bp signal in a manner consistent with established models for ORC/Cdc6 behavior in G1. Second, the *fkh1-R80A* allele caused substantial changes in the 80 ± 10 bp signals over the larger span representing all origins but not over the larger span representing all promoters. This observation provided evidence that the Fkh1-FHA domain was altering protein-DNA architecture substantially at origins but not promoters, consistent with Fkh1's prominent role at origins but accessory role in transcription. Here, interpretations of ORC/Cdc6-consistent signal were confined to the strongest signals coincident with and contiguous to the signal over the ORC site.

The nucleosome- and ORC/Cdc6-consistent signals were first examined at the FHA-ARS-dependent origin group (n = 16) [12] (Fig 7). In *FKH1* cells, the nucleosome signals 5' of the ORC site were reduced in G1-arrested relative to proliferating cells (Fig 7A). Furthermore, this nucleosome signal depletion was lost in the *fkh1-R80A* mutant (Fig 7B). Nucleosome depletion 3' of the ORC site, in contrast to nucleosome signal depletion 5' of the ORC site, did not show G1-arrest specificity. Nevertheless, this 3' depletion was also lost in the *fkh1-R80A* mutant. Thus, the relatively broad and shallow NDR of FHA-ARS-dependent origins observed under G1-arrest required the Fkh1-FHA domain. In contrast, while there was a prominent alteration in the origin-distal nucleosome signal in proliferating *fkh1-R80A* cells, the NDR itself was less dependent on the Fkh1-FHA domain under proliferating conditions (Fig 7C). In terms of ORC signals, virtually no differences were detected between G1-arrested and proliferating conditions (Fig 7D). However, the *fkh1-R80A* mutant showed a constriction of the ORC/Cdc6-consistent signal peak over the origin under G1-arrested but not proliferating conditions, reflecting the constriction of the NDR (Fig 7E and 7F).

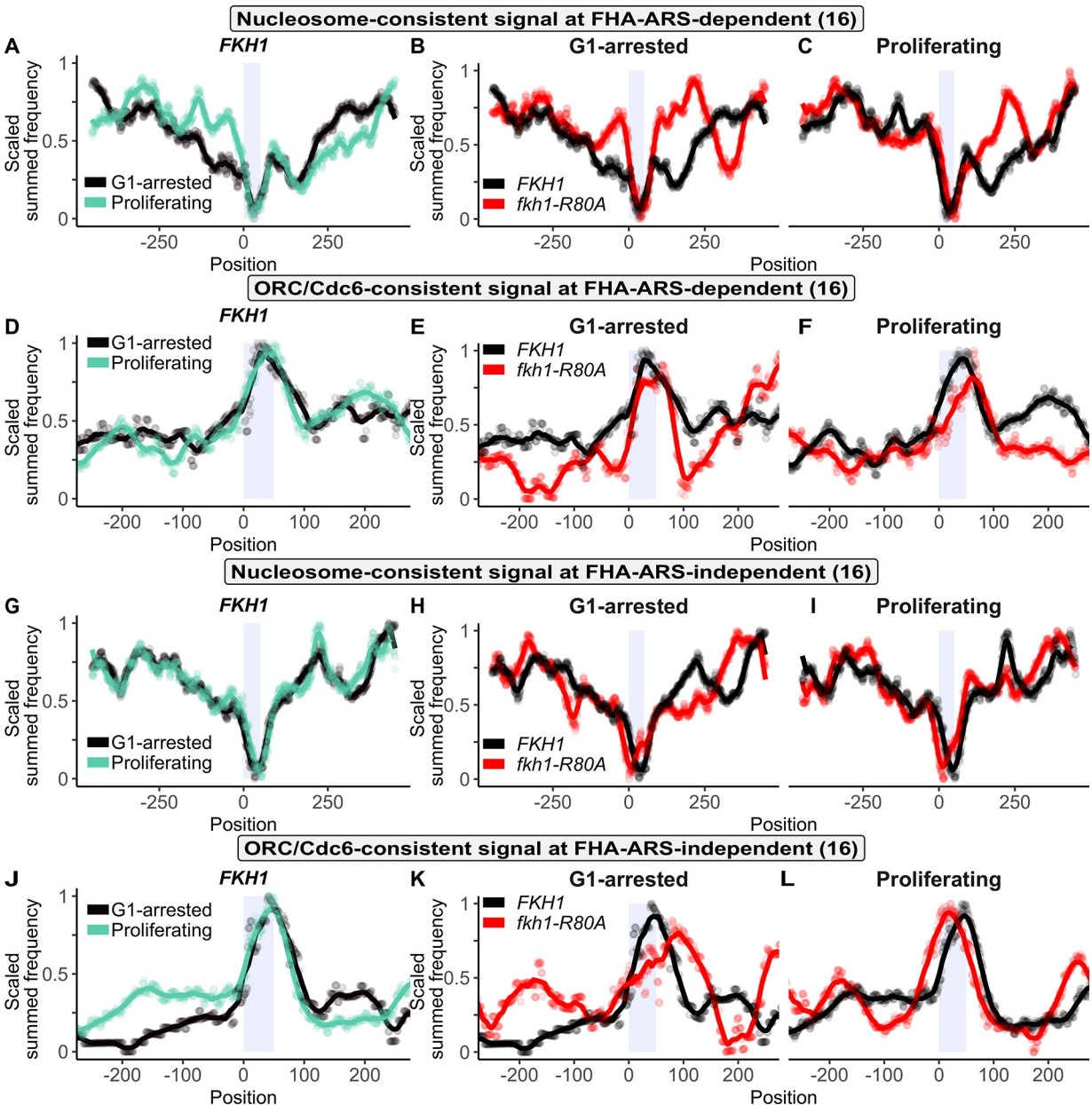

**Fig 7. The Fkh1-FHA domain promoted a broad NDR at FHA-ARS-dependent origins in G1-arrested cells.** Scaled nucleosome signals (generated from 8U MNase experiment) for: **(A)** The FHA-ARS-dependent origin cohort in the *FKH1* sample under G1-arrested and Proliferating conditions. **(B)** The FHA-ARS-dependent origin cohort in *FKH1* and *fkh1-R80A* samples under G1-arrested conditions. **(C)** The FHA-ARS-dependent origin cohort in *FKH1* and *fkh1-R80A* samples under Proliferating conditions. **(D-F)** As in (A-C), respectively, except for scaled ORC/Cdc6 signals. **(G-I)** As in (A-C), respectively, and **(J-L)**, as in (D-F), respectively, except for the FHA-ARS-independent cohort.

The protein-DNA architectural features noted for the FHA-ARS-dependent group were not observed for the contrasting control group of FHA-ARS-independent origins. First, the nucleosome signals under both G1-arrested and proliferating conditions for the FHA-ARS-independent group were similar to each other (compare Fig 7G–7A). Second, the NDR associated with FHA-ARS-independent origins was deeper, and relatively unaltered by the *fkh1-R80A* allele (Fig 7H and 7I). Third, the ORC/Cdc6-consistent signal over the origin was narrower than the analogous signal at FHA-ARS-dependent origins (compare Fig 7J–7D), and

broadened in the *fkh1-R80A* mutant under G1-arrested conditions, behaving oppositely to the ORC/Cdc6-consistent signal in response to *fkh1-R80A* at FHA-ARS-dependent origins. These higher-resolution analyses of the MNaseSeq data provided evidence that the Fkh1-FHA domain promoted a relatively broad, shallow NDR at the FHA-ARS-dependent origin group that showed G1-specificity.

## The Fkh1-FHA domain promoted an extended ORC/Cdc6-consistent signal at the FHA-SORT-positive origin group in G1-arrested cells

The approach used in Fig 7 was applied to the larger groups of FHA-SORT-regulated origins (Fig 8). First, the nucleosome- and ORC/Cdc6-consistent signals at the FHA-SORT-positive origins in *FKH1* cells were compared under G1-arrested and proliferating conditions (Fig 8A). For this group, nucleosome signals 3' of the ORC site were reduced in G1-arrested compared to proliferating cells. Thus, while the nucleosome signal profiles of both distinct groups of positively regulated origins, FHA-ARS-dependent and FHA-SORT-positive, were reduced in G1-arrested cells, the depletion of signal for FHA-ARS-dependent group occurred 5' of the ORC site while that for the FHA-SORT-positive occurred 3' of the ORC site (compare Figs 8A–7A). Under G1-arrested conditions, the *fkh1-R80A* mutant showed slightly enhanced nucleosome signals 3' of the ORC site, but the overall effect of this allele on the G1-arrested nucleosome profile was milder for FHA-SORT-positive origins compared to FHA-ARS-dependent origins (compare Figs 8B–7B). In proliferating cells, the *fkh1-R80A* had virtually no effect on the nucleosome signals (Fig 8C). Thus, in terms of the nucleosome signal profile, the FHA-ARS-dependent and FHA-SORT-positive groups differed, with the FHA-ARS-dependent group showing more substantial Fkh1-FHA-dependent effects in both G1-arrested and proliferating conditions. In terms of ORC/Cdc6-consistent signals, the FHA-SORT-positive origins showed substantial G1-arrest-specific and Fkh1-FHA-dependent changes (Fig 8D and 8E). In particular, the FHA-SORT-positive origin group showed a significant, extended 5' shoulder under G1-arrested conditions compared to either the FHA-ARS-dependent group (Fig 7A) or all origins (S9D Fig).

The particular nucleosome and ORC/Cdc6-consistent signals of FHA-SORT-positive origins were not observed for the contrasting control group of FHA-SORT-negative origins (Fig 8G–8L). For example, while the NDR for the FHA-SORT-positive group under G1-arrested conditions was slightly broadened on the 3' side of the ORC site relative to proliferating conditions, the NDR for the FHA-SORT-negative group was the same between these two growth conditions (compare Fig 8G–8A). In addition, while the NDR for the FHA-SORT-positive group was narrowed under G1-arrested conditions in the *fkh1-R80A* mutant, this mutant caused the opposite outcome at the FHA-SORT-negative group (compare Fig 8H–8B). However, the differences in the ORC/Cdc6 profiles between the two groups were the most striking. Specifically, the ORC/Cdc6 signal at the FHA-SORT-negative origins lacked the prominent G1-arrest specific, extended 5' shoulder of the FHA-SORT-positive group. Similarly, the ORC/Cdc6 signals for the FHA-SORT-negative origins over the origin in G1-arrested and proliferating conditions were virtually the same over the ORC site in both G1-arrested and proliferating conditions, in contrast to the ORC/Cdc6 signals over the FHA-SORT-positive group (compare Fig 8J–8D).

In summary, both nucleosome- and ORC/Cdc6-consistent signals as well as their relative responses to the *fkh1-R80A* mutant differed between FHA-SORT-positive and FHA-SORT-negative origins. In particular, analyses of the data from FHA-SORT-positive origins generated a G1-specific, expanded ORC/Cdc6-consistent signal that was not detected in the FHA-SORT-negative group.

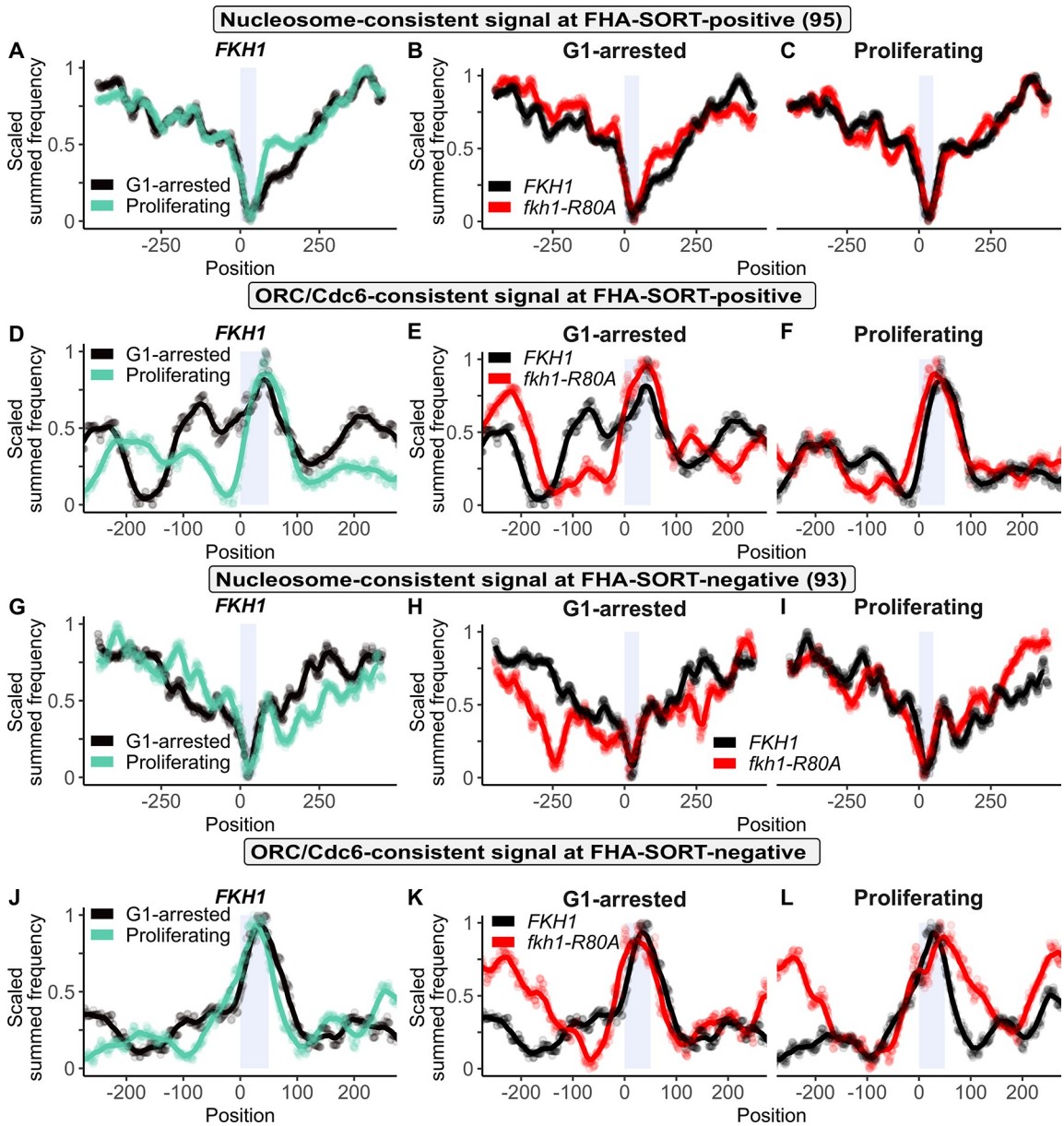

**Fig 8. The Fkh1-FHA domain promoted a 5' extended ORC signal at the FHA-SORT-positive origin group in G1-arrested cells.**
Scaled nucleosome signals (generated from 8U MNase experiment) for: **(A)** The FHA-SORT-positive origin cohort in the *FKH1* sample under G1-arrested and Proliferating conditions. **(B)** The FHA-SORT-positive origin cohort in *FKH1* and *fkh1-R80A* samples under G1-arrested conditions. **(C)** The FHA-SORT-positive origin cohort in *FKH1* and *fkh1-R80A* samples under Proliferating conditions. **(D-F)** As in (A-C), respectively, except for scaled ORC/Cdc6 signals. **(G-I)** As in (A-C), respectively, and **(J-L)**, as in (D-F), respectively, except for the FHA-SORT-negative origin cohort.

## Discussion

About 400 annotated origins distributed over 16 nuclear chromosomes promote the efficient and accurate duplication of the yeast genome. Multiple studies have focused on understanding the role of the forkhead proteins (Fkh1/2, Fkhs) in enhancing the activity of a substantial fraction of these origins, which include many origins that act in early S-phase. Fkhs' origin-stimulating role is viewed as direct, requiring that a Fkh protein bind origin-adjacent FKH sites to

enhance the probability that essential steps in the origin cycle will occur [5]. In contrast, Fkhs negatively regulate many late origins, but this negative role is viewed as indirect, wherein these late origins' enhanced activity in the absence of Fkh function is a byproduct of early origins' reduced ability to compete for limiting origin-control proteins [15,16]. Beyond this broad outline, the post-DNA binding role(s) of Fkhs in regulating origin activity are less clear [13,18,41]. Experimental evidence supports a role for Fkh1/2 at early origins in the direct recruitment of the limiting S-phase kinase, the DDK, that activates the loaded MCM complex, indicating that Fkh proteins stimulate the S-phase origin activation step [18]. There is also experimental evidence, including some data in this report, that support a role for the Fkh1 protein in the G1-phase origin licensing step [12,14,18]. Of course, Fkh1 could regulate both steps of the origin cycle, and to different extents at different origins. Regardless, definitive molecular details are unknown, including a consensus on the regions of Fkh1 most critical for its post-DNA binding function at origins.

Here, S-phase SortSeq experiments provided evidence that the Fkh1-FHA domain, and in particular its canonical pT-specific protein binding function, was responsible for regulating chromosomal origin activity on a genomic scale during a normal S-phase. Specifically, the Fkh1-FHA domain promoted the activity of most early origins and inhibited the activity of many late origins. In addition, MNaseSeq experiments were used to assess origin-associated protein-DNA architectural features. These experiments provided evidence that the Fkh1-FHA domain affected origin-associated chromatin hallmarks most strikingly in G1-phase. In particular, the Fkh1-FHA domain promoted extended ORC-consistent signals at positively regulated origins in G1-arrested cells, while having relatively minor effects on these origins in proliferating cells, and even opposing effects at negatively regulated origins. These and additional observations supported a model wherein the Fkh1-FHA domain promoted ORC binding and/or origin licensing at a substantial fraction of early origins.

## The Fkh1-FHA domain promoted the activity of most early S-phase origins, including centromere-associated origins

Two broad classes of origins were regulated by the Fkh1-FHA domain: FHA-SORT-positive origins were defined as those whose activity was reduced by the *fkh1-R80A* allele, while FHA-SORT-negative origins were defined as origins whose activity was enhanced. FHA-SORT-positive origins were comprised primarily of early origins, while FHA-SORT-negative origins were comprised of late origins. This general Fkh1-FHA origin-target profile was the same as the determined for Fkh1/2-regulated origins, where Fkh1/2-activated origins (positively regulated) are composed primarily of early origins and Fkh1/2-repressed origins (negatively regulated) are composed of late origins [10]. Fkh1/2-regulated origins are origins whose activity is altered in mutant yeast that lack the *FKH1* gene and contain a mutated and presumably less active version of the *FKH2* gene. Given these data, substantial overlap observed between the FHA-SORT-positive and Fkh1/2-activated origins made sense. However, it was surprising that the overlap between the FHA-SORT-negative and Fkh1/2-repressed groups was not significant. The intrinsically lower probability of activation of many late origins may account for the lack of overlap observed between these two independent and experimentally distinct origin studies. Alternatively, the distinct Fkh mutant backgrounds used in the two studies may be relevant to the composition of limiting origin regulatory proteins that are released from their early origin obligations, perhaps by altering undefined aspects of higher-order chromosomal neighborhoods (e.g. proximity to the inner nuclear membrane) or chromatin composition.

Because Fkh1 and Fkh2 can substantially substitute for one another, it was notable that a single missense mutation in *FKH1* altered the activity of so many yeast origins [6]. The

extensive sequence similarities between these forkhead paralogs, particularly over their conserved FHA and DNA binding domains, offers an obvious explanation that is also supported by Fkh1 and Fkh2 target binding studies [6,11]. In particular, Fkh2, in the absence of Fkh1, takes over many Fkh1 binding sites sufficiently enough that it is able to provide for the post-DNA binding functions of the missing Fkh. Thus, the observation that the *fkh1-R80A* allele reduced the activity of many of the origins initially defined as Fkh1/2-activated must mean that this mutant protein retained the ability to bind to Fkh1 target sites sufficiently enough to prevent Fkh2 from taking over these sites and providing FHA domain function. Mutational analyses of *ARS1529.5*, an origin that qualifies as FHA-ARS-dependent, FHA-SORT-positive and Fkh1/2-activated, supports this interpretation [12]. For this origin, the *fkh1-R80A* allele causes a more severe defect than a *fkh1Δ* allele, which in fact causes no defect at all. A parsimonious explanation is that the fkh1-R80A mutant protein retains the ability to bind to the required 5' FKH-T site in this origin, thus preventing Fkh2 from occupying this site and supplying FHA domain function.

The lack of more extensive overlap between the FHA-SORT-positive and Fkh1/2-activated origin groups was not surprising given the substantial differences in both genetic backgrounds and methods used to identify these origin collections. In particular, several studies have implicated regions outside of the FHA domain required for Fkh origin control [11,13,18]. However, the identification of CEN-associated origins as FHA-SORT-positive might point to potentially relevant differences between the experimental methods used to define FHA-SORT-regulated origins and Fkh1/2-activated origins. Specifically, CEN-associated origins are among the earliest origins in yeast, yet do not qualify as Fkh1/2-activated. Instead, CEN-associated origins depend on a kinetochore protein to recruit Dbf4 [17]. In SortSeq experiments, origin activity data are derived from S-phase cells that are harvested from a proliferating cell population. In the BrdU-IP method used to identify Fkh1/2-activated origins, origin activity data are derived from cells progressing through a synchronized S-phase, typically under conditions that deplete dNTPs and activate the Rad53 cell cycle checkpoint, retarding replication fork progression and late origin activation [42–44]. Specifically, early origins initiate DNA synthesis and incorporate dNTPs (and thus BrdU) only over relatively short DNA spans before replication forks stall. Fkh1/2-activated origins are defined by their failure to incorporate BrdU nucleotides efficiently under these conditions in forkhead mutants compared to wild-type controls, presumably because their activity has been delayed enough that they now respond to the Rad53 checkpoint. Therefore, a possible explanation for the identification of CEN-associated origins as FHA-SORT-positive but not Fkh1/2-activated is that the latter origins act so early that they escape the Rad53-checkpoint, perhaps because they are still firing earlier than most origins even in a forkhead mutant.

## Global and origin-specific roles for the Fkh1-FHA domain in G1-arrested chromatin

Distinct chromatin hallmarks help define origins. These hallmarks include a nucleosome-depleted region (NDR) flanked by an array of relatively positioned nucleosomes [32]. These chromatin features are interdependent, and ORC-origin binding as well as one of several distinct ATP-dependent chromatin remodelers are sufficient to establish these hallmarks *in vitro* [33]. To examine whether the Fkh1-FHA domain altered these origin-associated chromatin features, MNaseSeq experiments were performed on *FKH1* and *fkh1-R80A* cells under G1-arrested and proliferating conditions. Lower-resolution analyses of these data revealed that the Fkh1-FHA domain did not have a major impact on nucleosome positioning or the NDR. Thus, while the *fkh1-R80A* mutant altered ORC binding at many origins, this effect was not

sufficient to cause the collapse of nucleosome positioning associated with a complete loss of ORC-origin binding. Instead, these analyses revealed an unexpected phenotype. Specifically, in *fkh1-R80A* mutants, nucleosome stability, defined here as the nucleosome to subnucleosome ratio, was altered. In G1-arrested cells, nucleosome stability was reduced by the *fkh1-R80A* allele, while in proliferating cells, nucleosome stability was enhanced. No link was detected between the magnitude of these effects and Fkh1-FHA-regulated origin activity. Moreover, the phenomenon was also observed at promoter-adjacent nucleosomes. Thus, the Fkh1-FHA domain was required for the normal stability at loci characterized for their highly-ordered nucleosome arrays. Because Fkh1 is a transcription factor, it is possible that changes in the expression of undefined factor(s) have altered the yeast cell such that the levels of MNase able to access chromatin under the experimental conditions has also been altered. Additional experiments are required to decipher the mechanistic basis of this unexpected phenomenon.

Higher-resolution analyses of normalized MNaseSeq data were used to explore the effect of the Fkh1-FHA-domain on origin-associated protein-DNA architecture and ask whether any Fkh1-FHA-dependent structural features could be linked to Fkh1-FHA-dependent origin activity. In contrast to the low resolution analyses, these higher-resolution analyses provided evidence that the Fkh1-FHA domain promoted structurally distinct origin-associated chromatin architectures at the two groups of positively regulated origins. For the small FHA-ARS-dependent group, the most dramatic Fkh1-FHA-dependent protein-DNA architectural feature was a relatively shallow, broad NDR under G1-arrested conditions. Of particular note was the G1-arrest-specific Fkh1-FHA-dependent depletion of nucleosome signal 5' of the ORC site. Given the functional link between a FKH motif in the vicinity of this position (i.e. 5' FKH-T), a distinctly enriched feature of this origin group, this outcome was particularly intriguing. Speculatively, perhaps Fkh1 binding to these motifs helps recruit chromatin remodelers that contribute to these origins' distinct NDRs and facilitate ORC binding and/or subsequent events in G1-phase origin licensing. However, the Fkh1-FHA domain also contributed to nucleosome signal depletion 3' of the ORC site, even though this particular group of origins was distinct for its relative depletion of 3' FKH-A motifs. Nevertheless, for this group, the requirement for the 5' FKH-T site in Fkh1-FHA-dependent ARS activity has been experimentally verified, so that a nucleosome change juxtaposed near these sites warrants future study. MNaseSeq experiments to analyze nucleosome signals at individual origins within this group and the effect that specific FKH site mutations have on these signals are important future experiments.

The much larger collection of FHA-SORT-positive origins did not show the same FHA-dependent nucleosome signal behavior as the FHA-ARS-dependent group. Of course, FHA-SORT-positive origins were defined solely based on the effect of the Fkh1-FHA-domain on origin activity, while the much smaller FHA-ARS-dependent origin group was defined based on a relatively stringent definition of ORC-origin binding. Thus, the FHA-SORT-positive group was likely to be represented by a more mechanistically diverse collection of origins than the FHA-ARS-dependent group. Nevertheless, the FHA-SORT-positive group also showed a striking G1-specific, Fkh1-FHA-dependent protein-DNA architecture. For this group, the most striking architectural feature was observed for ORC signals, and particularly a dominant shoulder 5' of the major ORC site that was lost in the *fkh1-R80A* mutant cells. These observations were consistent with the high-resolution ORC ChIPSeq data that also showed a Fkh1-FHA-dependent ORC signal 5' of the ORC site. Based on what is known about ORC's role at origins during G1-phase, a reasonable explanation for this 5' extended ORC signal was that many or most origins in the FHA-SORT-positive group efficiently formed stable ORC-Cdc6 complexes compared to origins in the control FHA-SORT-negative group.

However, early origins, which dominate the FHA-SORT-positive group, also recruit MCM complex activation factors in late G1, prior to S-phase, and these factors and possibly pre-activation remodeling of the MCM complex or release of the origin from the inner nuclear membrane might also alter ORC-DNA interactions [14]. Regardless, these chromatin analyses reinforced the idea that Fkh1 plays its role at origins during G1-phase, prior to origin firing, and extended it by showing that the Fkh1-FHA domain, and by inference as yet undefined pT-protein partner(s), are key to establishing distinct and G1-arrest-specific protein-DNA architectural features associated with Fkh1-FHA-dependent origins.

## Methods

### Construction of yeast strains

The yeast strains used were congenic derivatives of W303, and the *fkh1-R80A* mutation was introduced into the native *FKH1* locus using standard procedures [22]. Each strain shared the following genotype: *MATa*, *ADE2*, *lys2Δ*, *his3-11*, *leu2-3,112*, *trp1-1*, *ura3-1*, *RAD5*. Three wild-type (*FKH1*) strains (CFY3533, CFY3534, KPY448) and two mutant (*fkh1-R80A*) strains (CFY3956, CFY3957), isolated as independent segregants in genetic crosses, were used for the experiments in this study.

### SortSeq experiments

Yeast were grown at 30˚C and processed as described in [25,27] except that a Sony MA900 was used to separate and collect S-phase and G2-phase cells (S1 Fig for cell-cycle histograms and gates). Genomic DNA was isolated as described [25]. Libraries for one wild-type sample were prepared at Northwestern University and single-end sequenced. The remaining libraries were paired-end sequenced in-house using a P2-100 flow cell and a NextSeq1000. The libraries for in-house sequencing were made using the 1/10 dilution of the adaptor and amplified for seven cycles using the NEBNext Ultra II kit. Sequencing reads were mapped to the yeast genome, build sacCer3, using Bowtie2 [45] with the following options: bowtie2 -p 4 -X 750 -q–phred33. Mapped reads were converted to single nucleotide read coverages using Bedtools genomecov [46]. For the *FKH1* libraries sequenced at Northwestern (SRR27983881(*FKH1*-3-G2) and SRR27983882 (*FKH1-3*-S)), there were >30 million uniquely genome-aligned reads. For the libraries sequenced in-house, there were > 5 million uniquely aligned reads (SRR27983878 (*fkh1-R80A*-2-G2), SRR27983879 (*fkh1-R80A*-2-S), SRR27983880, (*fkh1-R80A*-1-S), SRR27983883 (*FKH1*-2-S), SRR27983894 (*FKH1*-1-G2), SRR27983895 (*FKH1*-1-S). Single nucleotide coverages were normalized for sequencing depth and breadth [31] then summarized into 1 kbp bins. S/G2 ratios were determined for each independent cell population (three independent populations of *FKH1* cells and two independent populations of *fkh1-R80A* cells) using the 1 kbp bin values. Outlier S/G2 ratios were defined as any ratio that was over 1.5 IQRs below the first quartile (Q1) or above the third quartile (Q3) and were removed. The resulting set of ratios were next scaled to be between 1 and 2 using the following equation: scaled ratio = (ratio—min(ratio))/(max(ratio)—min(ratio)) + 1. The resulting scaled ratios were smoothed by fitting data to a cubic smoothing spline (R function: smooth.spline, default parameters). Smoothed data from a given genotype (three samples for *FKH1* and two samples for *fkh1-R80A*) were statistically summarized by finding the means and 95% confidence intervals.

### FKH1 motif mapping

The FKH1 consensus motif was determined and challenged previously [12]. Briefly, the program ChIPOTle (parameters of 400 and 100 bp window and step size, respectively) was used

to identify Fkh1 ChIPSeq peaks meeting -log10 P-value $\geq$ 3 [47]. The motif finder tool in MochiView was used to identify motifs enriched within these Fkh1 peaks, specifying an 8 bp motif and using a 4th order Hidden Markov Model of the entire genome.

## ORC ChIPSeq

These wet-bench experiments were performed as described in [12]. The raw data can be accessed at BioProject PRJNA694026.

## MNaseSeq

Procedures were as in [33]. Briefly, a 500 ml culture in liquid YPD was incubated at 30°C for 18 hours until $A_{600}$ of 0.6 ~ 1.0. The cells were collected and nuclei were isolated. After spheroplast generation (incubation with zymolyase (MP Biomedicals)), cells were washed and resuspended in a Ficoll solution (18% Ficoll Type 400 (Sigma-Aldrich), 1 mM $MgCl_2$, 20 mM $KHPO_4$, 0.25 mM EGTA pH 8.0, 0.25 mM EDTA pH 8.0). Spheroplasts were aliquoted and centrifuged for 30 min and the nuclei were flash frozen in liquid $N_2$ and stored at -80°C. Nuclear chromatin was digested with either 8 or 16 units of micrococcal nuclease (MNase) (Sigma-Aldrich) in 150 mM Tris-HCl pH 7.5, 500 mM NaCl, 14 mM $CaCl_2$, 2 mM EDTA, 2 mM EGTA and 50 mM $\beta$-mercaptoethanol to enrich for mononucleosomes (80% mono, and 20% dinucleosomes). After the reaction was stopped (by adding 10 mM EDTA and 0.5% SDS), the DNA was purified by proteinase K digestion, phenol-chloroform extraction, ethanol precipitation, RNAse A digestion and isopropanol precipitation. After running a 1.5% agarose gel, the mononucleosomal DNA was enriched by gel extraction by the PureLink Quick Gel Extraction kit (Invitrogen). Sequencing libraries were prepared as in [25] with the NEBNext Ultra II DNA Library Prep Kit for Illumina (NEB E7645S). Reads from all sequencing experiments were annotated against the yeast genome, sacCer3 build, using Bowtie2 and the following parameters: -p 8 -X 250 -q–phred33 –no-discordant–no-mixed–no-unal. Thus annotated, reads were converted to 2D occupancy plots (frequency of MNase protections as a function of position within user-defined loci) at collections of origins and genes through the software plot2DO [35]. 2D occupancy plots were analyzed as outlined in Figs 2, 6A and S8.

## Supporting information

**S1 Fig. Defining the S-phase and G2-phase populations used to generate the SortSeq data.** The cell density histograms from the proliferating yeast population as a function of fluorescence are shown for **(A)** Three independent *FKH1* yeast and **(B)** Two independent *fkh1-R80A*. Vertical dotted lines indicate the gates used for the collection of the S-phase and G2-phase populations that were processed for sequencing.
(TIFF)

**S2 Fig. S-phase SortSeq scans for all yeast nuclear chromosomes.** Please see text and Figure legend sections relevant to main text Fig 1A.
(TIFF)

**S3 Fig. Origin-by-origin assessment of the SortSeq data generated by the FHA-ARS-regulated origin groups. (A)** Ratio of ARS activity in *fkh1-R80A* cells to *FKH1* cells (y-axis) for indicated origins (xaxis). FHA-ARS-dependent origins indicated by black bars while FHA-ARS-independent origins indicated by grey bars. **(B)** Close-up of S-phase copy number scans for FHA-ARS-regulated origins in *FKH1* (black) and *fkh1-R80A* (red) cells. **(C)** The *fkh1-R80A/FKH1* S-phase copy number ratio determined for each FHA-ARS regulated origin color coded based on its FHA-SORT assigned status. FHA-ARS-regulated origins were defined

initially based on ORC-origin binding mechanisms and subsequently parsed by ARS activity. Neither *ARS609* nor *ARS224*, each an FHA-ARS-dependent origin, qualified as FHA-SORT-positive. Note that neither origin is active even in *FKH1* cells on the chromosome. Thus ORC-origin binding mechanisms might not play a major role in their chromosomal activity, yet might have a large impact on origin activity in a more isolated situation, such as on a plasmid. (TIFF)

**S4 Fig. Normalized S-phase copy number scans across yeast centromeres.** Based on the stringent definition of Cen-associated origins used here, CEN4, CEN6, CEN7, and CEN15 do not contain CEN-associated origins per our definition. Copy numbers are presented as means and 95% confidence intervals from three *FKH1* (black) and two *fkh1-R80A* (red) replicates, as discussed in the main text.
(TIFF)

**S5 Fig. Analyses of FKH1 motifs at FHA-SORT-positive Cen-associated origins.** The analyses in Fig 4, main text, were applied to Cen-associated origins that were FHA positive. **(A)** The fraction (y-axis) of origins within the indicated group (x-axis) containing at least one match to the indicated motif within the indicated origin regions for the relevant FHA-regulated origin groups classified by SortSeq. "5' FKH1-T" queried nucleotides -150 through -11 for FKH1-T motif matches, while "3' FKH1-A" queried nucleotides +41 through +150 for FKH1-A matches. The "ORC-overlapping" region encompassed nucleotides -10 through +40 and was queried for FKH1-T motifs. The horizontal line indicates the fraction of all confirmed origins (n = 393) that contained a match to the queried FKH motif in the origin region under assessment. The enrichment or depletion of a given motif in any given origin group was challenged against the fraction of that motif in all confirmed origins using the hypergeometric distribution function. Significant P-values are denoted by asterisks (*, $P < 0.05$; **, $P < 0.001$; ***, $P < 0.0001$). In these analyses, 150 bp regions 5' and 3' of the ORC site were queried. **(B)** The cumulative fraction of origins (y-axis) in the indicated groups containing a 5'FKH-T (left) or a 3'FKH-A (right) after traversing the indicated number of nucleotides from the ORC site (x-axis). Nucleotide positions that reached P-value significance values of 0.01 are indicated by gray cross marks derived from hypergeometric distributions where at each position, the fraction of origins in the queried collection that contained a match by that nucleotide position was reached is compared to the fraction of all confirmed origins (n = 393) that contained a match by the same position. **(C)** Summary of the 50% accumulation point for the analyses in (B).
(TIFF)

**S6 Fig. Summarized internally scaled ORC binding levels at three types of origins.** Data are as presented in Fig 5A, 5D, and 5G, but with the inclusion of 95% confidence intervals indicating level of variation within the group being graphed.
(TIFF)

**S7 Fig. The Fkh1-FHA domain promoted nucleosome stability similarly at FHA-ARS-dependent and FHA-ARS-independent origin groups.** The analysis of MNaseSeq data for all origins described in Fig 6 were applied to the contrasting groups of FHA-ARS-regulated origins as defined in Fig 2A.
(TIFF)

**S8 Fig. The Fkh1-FHA domain promoted nucleosome stability similarly at FHA-SORT-positiveand FHA-SORT-negative origin groups.** The analyses of MNaseSeq data for all origins described in Fig 6 were applied to the contrasting groups of FHA-SORT-regulated origins

as defined in Figs 1 and 2B.
(TIFF)

**S9 Fig.** Nucleotide-resolution, scaled nucleosome (A, B) or ORC (C, D) signals for all 393 origins or all 6000 promoters in *FKH1* and *fkh1-R80A* cells under G1-arrested or Proliferating conditions, as indicated. The arrow in panel C indicates a G1-arrest specific and Fkh1-FHA-dependent prominent 5'shoulder on the ORC-signal peak.
(TIFF)

**S10 Fig. The Fkh1-FHA domain suppressed nucleosome signals at FHA-ARS-dependent origins and promoted an extended ORC signal at the FHA-SORT-positive origins in G1-arrested cells.** Scaled single-nucleotide signals generated from the 8 U MNase experiment for *FKH1* and two independent *fkh1-R80A* mutant isolates for **(A)** FHA-ARS-dependent cohort nucleosome signals **(B)** FHA-ARS-dependent cohort ORC signals **(C)** FHA-SORT-Positive cohort nucleosome signals **(D)** FHA-SORT-Positive cohort ORC signals.
(TIFF)

## Acknowledgments

We thank members of the laboratory and the Department of Biomolecular Chemistry for support and discussions. We are particularly grateful to Christina Hull and her lab. We also thank Pyush Lai and the laboratory of Patricia J Kiley (Department of Biomolecular Chemistry and the Great Lakes Bioenergy Research Center) for sharing their Sony MA900, and training us in its use and care. We are extremely grateful to Conrad Nieduszynski (The Earlham Institute) for sharing data with us and his advice on SortSeq experiments and analyses.

## Author Contributions

**Conceptualization:** Timothy Hoggard.

**Data curation:** Timothy Hoggard.

**Funding acquisition:** Christoph F. Kurat, Catherine A. Fox.

**Investigation:** Timothy Hoggard, Erika Chacin.

**Methodology:** Timothy Hoggard, Erika Chacin, Christoph F. Kurat.

**Project administration:** Christoph F. Kurat, Catherine A. Fox.

**Resources:** Erika Chacin, Allison J. Hollatz.

**Supervision:** Christoph F. Kurat, Catherine A. Fox.

**Validation:** Timothy Hoggard, Erika Chacin.

**Visualization:** Timothy Hoggard, Catherine A. Fox.

**Writing – original draft:** Timothy Hoggard, Catherine A. Fox.

**Writing – review & editing:** Timothy Hoggard, Allison J. Hollatz, Catherine A. Fox.

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
