## [Decision Letter · Decision Letter 0]

4 Apr 2024

Dear Catherine,

Your manuscript has been reviewed by three experts in the field.  Although all three agree that the work is interesting and appropriate for PLOS Genetics, Reviewer 3 raises some substantive technical issues that would need to be addressed before manuscript could be considered for publication.  The standard time frame for resubmission is 60 days.  However, if the necessary revisions require more time than that, extensions are available.  Please feel free to contact me with any questions.

Nick

Boilerplate follows:

Thank you very much for submitting your Research Article entitled 'The budding yeast Fkh1 Forkhead associated (FHA) domain promotes the G1-chromatin status and activity of chromosomal DNA replication origins' to PLOS Genetics.

The manuscript was fully evaluated at the editorial level and by independent peer reviewers. The reviewers appreciated the attention to an important problem, but raised some substantial concerns about the current manuscript. Based on the reviews, we will not be able to accept this version of the manuscript, but we would be willing to review a much-revised version. We cannot, of course, promise publication at that time.

If you decide to revise the manuscript for further consideration at PLOS Genetics, please aim to resubmit within the next 60 days, unless it will take extra time to address the concerns of the reviewers, in which case we would appreciate an expected resubmission date by email to plosgenetics@plos.org.

We are sorry that we cannot be more positive about your manuscript at this stage. Please do not hesitate to contact us if you have any concerns or questions.

Yours sincerely,

Nicholas Rhind

Guest Editor

PLOS Genetics

Gregory P. Copenhaver

Editor-in-Chief

PLOS Genetics

Reviewer's Responses to Questions

**Comments to the Authors:**

Reviewer #1:

Hoggard et al. have used a combination of genome-wide assays to address the role of FHA domain (R80A mutant vs. WT) of budding yeast Fkh1 in chromosome DNA replication. Sortseq reveals that FHA stimulates 95 early origins including periCEN origins and inhibits 93 late origins in S-phase. ORC CHIP-seq data show FHA promotes ORC-early origin binding, and MNase-Seq show FHA regulates nucleosome configurations in G1 phase. The authors suggest that post binding to early origins, Fkh1-FHA shapes chromatin architecture in G1 phase, which determines the origin activity in S phase.

Major concerns:

1. Fig1. Why different replicates were used for FKH1 (n = 3, black) and fkh1-80A (n = 2, red)? Did the authors confirm their large-scale seq data by qPCR at representative origins (i.e., peri-CEN ones) or other different approaches?

2. Fig3A. the authors compared FHA-SORT-positive based on fkh1-R80A with Fkh1/2-activated ori. Fkh2 also contains a FHA domain and plays a redundant role in origin activation with Fkh1, is Fkh2-FHA also involved as Fkh1-FHA? What’s the effect of double FHA mutants? The relatively lower (25/93) negative implicates that it may be worth to examine double FHA mutants, although the reader agree about the dominant negative effect of R80A.

3. Fig3D. The most interesting result is 12 cen-associated origins are affected in R80A but not in fkh1/2 mutants. These CEN-ori have Fkh1 ChIP signals. Have they tried to delete FKH motifs from one or two CEN-ori as their previous work to validate this important result? Are the other 5 CEN-ori showing FHA-SORT-neg due to lacking FKH motifs? If so, inserting proper FKH motifs to one of the 5 CEN-ori (FHA-SORT-neg) to see if it is sufficient to make it FHA-dependent.

4. Fig4. A related question to above, the characteristics of 5’ FKH1-T or FKH1-A motif in CEN-ori are missing in Fig4.

5. Fig5-8. How does FHA-stimulated ORC-ori binding affect nucleosome configurations? An alternative possibility is that it promotes more MCM loading, which is also known for early replication in budding yeast.

6. As already mentioned in their Discussion, for a deeper understanding of Fkh in origin activation, it will be of great interest to use the genome-wide approaches in this study to compare with other separation-of-function mutants, e.g., dimerization or DDK interacting, particularly combining these mutations with FHA in future.

Reviewer #2:

Here the authors further refine the relationship between the FHA domain of Fkh1 and its relationship to replication activity, nucleosome organization and ORC/Mcm binding. This work builds on their prior work, but the sort-seq and chromatin experiments provide important new insights into how individual classes of origins are regulated.

Major points:

I worry that the chromatin data might be over-digested and may not be preserving the ORC footprint at the origins. It would be nice to show an ethidium bromide stained gel of the nucleosome ladders used for the analysis to facilitate comparison and explain the differences in the patterns with prior studies. For example, the MacAlpine group typically performs a limited-digest with MNase such that 4-5 rungs of the nucleosomal ladder are resolvable on an ethidium bromide stained gel to preserve ORC and TF footprints.

The generation subnucleosomal complexes from (presumably over-digestion) was very interesting and suggest that there might be differential nucleosome accessibility/breathing at origins that is FHA dependent. This may be an opportunity for a more focused analysis on the distribution of nucleosomes and different subnucleosome complexes (not just in aggregate), but a breakdown of the individual species for each flanking nucleosome. I think there might be some insightful patterns that were missed in the bulk analysis captured by the teal and black boxes.

ORC footprints were almost non-existent in the raw chromatin data prior to "positional probability" normalization as described in (Figure S8). I just don't see how the authors got from the middle panel to the third panel as there is a negative inflection in the ORC summation data in the first panel that somehow becomes a peak in the third panel.

I struggled with Figures 7 and 8 and I am concerned that some of the ORC specific differences might be driven by an outlier or two especially where the number of origins in the set was limited. Is it possible to put confidence intervals on the curves? Or present the positional probability data as heatmap (with each row representing an origin)?

The upstream 'double ORC' peak that is in FHA positive sort and Fkh1 dependent (Figure 8B) -- could the upstream peak represent a Fkh1 footprint?

Reviewer #3: 

In this manuscript, Hoggard et al use Sort-seq to compare the dynamics of DNA replication in cells with and without a functional FHA domain in Fkh1 – a protein whose role in DNA replication initiation has been well studied by several groups including the Fox lab. The central claims of the manuscript are that the FKH1 FHA domain promotes firing of ~100 origins while inhibiting the firing of a similar number, and that the FHA domain promotes normal ORC binding and chromatin structure at origins.

The regulation of origin firing by forkhead is an interesting topic, and the conclusions here could be of interest to those interested in the dynamics of DNA replication. However, I have a number of significant concerns about the analyses presented and, more importantly, the data underlying them. These issues would need to be fully addressed in order for me to be willing to consider a revised version of this manuscript.

As shown in Figure 1 and discussed in the text, the Sort-seq data are noisy and relatively inconsistent between replicates. The authors suggest in the text that stochastic origin use likely underlies this variability, but Sort-seq analyses generally uses millions of cells (30M in the original description of the method by the Nieduszynski lab, down to ~1M for the lowest-input analysis of which I am aware). Across such a large number of cells, variability due to stochastic origin use should smooth out to give much more reproducible profiles. A more likely scenario in my opinion is variability in gating, such that cells in early or late S-phase are under- or over-represented. It is very to glean meaningful information about gating from the low-resolution data provided in Figure S1.

There are also possible coverage issues, as evidenced by the sharp peaks at ~200kb and ~750kb in Figure 1A (similar issues are also visible in Figures 1B, 2B and 3A). Such sharp transitions in coverage in some but not all datasets suggest that there might be problems with the quality of input material – if this were an analysis issue one would expect similar problems to manifest across all datasets. The sequence coverage reflected in the raw data files indeed appears to be insufficient for Sort-seq (considering data loss due to rDNA, mtDNA and PCR duplicates)

Based on the data deposited in the SRA, the GC content of the sequenced DNA is a long way from that expected for S. cerevisiae (e.g. SRR27983880 has 54.8% GC; SRR27983878 52.8% GC, and SRR27983879 is 53.8% GC, while SRR27983882 is a more reasonable 39.6% GC, mirroring the actual GC content of the yeast genome). I find this discrepancy concerning, and speculate that it may represent a substantial amplification bias.

The points above make it hard for me to draw firm conclusions from much of the downstream analysis, so I will keep my comments on these relatively brief. However, my major concerns in this regard are as follows.

1. Is it possible, using Sort-seq, to deconvolve origin firing efficiency vs firing time? A change in either could give rise to a similar Sort-seq profile, but these are distinct behaviors.

2. Some of the analysis feels over-worked from my perspective. For example, the sequencing data in Figure 4 suggest that there is a large ORC peak immediately upstream of the origin in FKH1 cells, which is lost in fkh1-R80A. The interpretation (or at least, my interpretation of the interpretation) is that there is a change in ORC positioning rather than the simpler explanation than loss of a specific peak while the rest of the region remains relatively unchanged.

3. On a similar theme, the analyses in Figures 7 and 8 appear to have normalized such that the minimum and maximum values within the analyzed range are 0 and 1, respectively. I suspect that this normalization has a big impact on the interpretation. Considering Figure 7F, if the effect of the FHA mutation is to essentially abolish the ORC peak, appropriately normalized data for ORC coverage would almost certainly look similar between WT and mutant outside the peak region, with the absence of a peak being the major difference. Artificially expanding the y-axis by normalizing to the maximum value in the range makes it look as though new ORC occupancy peaks appear in the mutant. As an additional note here, the data presentation in Figures 7 and 8 is so busy that it’s almost impossible to take anything away without extreme effort on the part of the reader.

4. The data in figure 6 suggest that the FHA domain stabilizes nucleosomes in G1 and destabilizes them in other phases of the cell cycle (based on the consistent observation that nucleosome:sub-nucleosome ratios are lower in G1 cells and somewhat higher in cycling cells, which are a mixed population that includes G1). This effect is observed at TSS and origins – i.e. essentially all well-positioned nucleosomes. The authors appear to disregard the apparently increased nucleosome stability in cycling cells. I am surprised that the strong likelihood of this being an indirect effect is only briefly considered: these data seem like a stray observation to me.

5. Various aspects of the methods, including the Sort-seq, are inadequately described. It is also not clear to me why the authors apparently did not use pre-existing Sort-seq data analysis pipelines

**Have all data underlying the figures and results presented in the manuscript been provided?**

Reviewer #1: Yes

Reviewer #2: Yes

Reviewer #3: Yes

PLOS authors have the option to publish the peer review history of their article (what does this mean?). If published, this will include your full peer review and any attached files.

<b>Do you want your identity to be public fo

---

## [Decision Letter · Decision Letter 1]

9 Jul 2024

Dear Catherine,

We are pleased to inform you that your manuscript entitled "The budding yeast Fkh1 Forkhead associated (FHA) domain promotes the G1-chromatin state and activity of chromosomal DNA replication origins" has been editorially accepted for publication in PLOS Genetics. Congratulations!

Note that Reviewer 3 has provided a number of thoughts on the reproducibility of your replication profiles, which you can in corporate into your final version, at your discretion.  Likewise, Reviewer 2 recommends publishing your review history, which is also up to you.

Yours sincerely,

Nick

Nicholas Rhind

Guest Editor

PLOS Genetics

Gregory P. Copenhaver

Section Editor

PLOS Genetics

Comments from the reviewers (if applicable):

Reviewer's Responses to Questions

**Comments to the Authors:**

Reviewer #1: The authors have addressed the main concerns properly. I support the publication.

Reviewer #2: I am satisfied with the detailed response from the authors. I would also encourage the authors to consider publishing the reviews as the authors provided a lot of context and detailed expert knowledge on these different class of origins that would be useful for the field and might be missed in a cursory read of the paper.

Reviewer #3: Having replicates is certainly beneficial for validating the accuracy of the sort-seq profiles. However, I still question their robustness and their impact on downstream analysis. Replication timing is highly consistent in S. cerevisiae. For instance, in Müller et al. (2014), the timing profiles generated in WT haploid and diploid cells using sort-seq are almost identical, despite originating from different cell lines. This consistency is also observed when comparing sync-seq and sort-seq profiles from different samples. Similarly, in Natsume et al. (2013), except at centromeres, timing profiles are identical between WT and DB4-myc cells across the rest of the genome. Very recently, in a pre-print introducing the Nanotiming technique, Theulot and colleagues generated multiple replicates for the same genotype in WT and mutant cells, showing almost perfect identity for a given genotype. Therefore, there is no intrinsic reason to expect significant variations between the replicates presented here, but rather a near-perfect overlap for a given genotype.

Sequencing depth could explain the lack of consistency in the profiles based on the results presented here. For two WT replicates and the two fkh1-R80A sort-seq experiments, about 5 million reads are uniquely mapped, corresponding to about 450 reads per kb. Although this is sufficient to generate a rough timing profile, it is around half the number required for an accurate profile computed at 1kb windows. As Batrakou et al. (2020) state, "We find that a depth of 1,000–2,000 reads/bin gives cost-efficient sequencing depth without compromising the precision of the data."

The impact of rDNA copy number variation could be a partial explanation for variation. However, systematic variation seems unlikely to occur in WT cells across five isogenic clones without selective pressure. If variations in copy number only occur in fkh1-R80A cells, this would suggest a role for FKH1 in the replication timing of the rDNA locus, which has never been described to my knowledge. Differences in culture conditions (such as media or temperature) are more likely to result in changes in S-phase duration and the absolute timing of an origin, but not in relative timing, which is what sort-seq captures.

I appreciate the additional data from Conrad Nieduszynski, which indeed confirms the main conclusion. I am not questioning the existence of this allele's effect, but rather the precision of the number of origins detected as affected, which, as the authors themselves state, appears to be "underestimated." For some origins, changes in activation timing appear to be quite subtle and thus require absolute confidence in the data. The differences presented between WT profiles generated by the authors in RFig1 and RFig2, and between their profiles and Conrad Nieduszynski’s profiles in RFig3 and RFig4, although showing the same trend for a given genotype, do not fully alleviate my concerns about the robustness of the sort-seq experiments presented in the manuscript. I have no concerns regarding the downstream analysis pipeline (chromosome scan and KDE analysis), which is further validated by the results involving mph1-2TA in RFig7. Additionally, I agree that the definition of appropriate smoothing is always a topic of debate.

I find the differences in the KDE analysis between the authors' data and Conrad Nieduszynski’s data presented in RFig5 and RFig6 striking. Although there is significant overlap, only 44% of the FHA-positive origins overlap between the two datasets according to RFig6. This discrepancy illustrates some lack of robustness in the generated sort-seq profile. This lack of overlap may also explain why the overlap between the FHA-SORT-negative and Fkh1/2-repressed origins is not significant in the study.

Ultimately, while I still have reservations about data quality, I agree with the authors' statement that the noise observed is not sufficient to explain the observed effect of the fkh1-R80A allele and thus the main conclusions of the study. The authors may be missing some interesting biology as a result of sparse data, but all claims in the manuscript are supported.

**Have all data underlying the figures and results presented in the manuscript been provided?**

Reviewer #1: Yes

Reviewer #2: Yes

Reviewer #3: Yes

PLOS authors have the option to publish the peer review history of their article (what does this mean?). If published, this will include your full peer review and any attached files.

Reviewer #1: No

Reviewer #2: No

Reviewer #3: No

**Data Deposition**

http://datadryad.org/submit?journalID=pgenetics&manu=PGENETICS-D-24-00205R1

**Press Queries**

---

## [Editor Report · Acceptance letter]

31 Jul 2024

PGENETICS-D-24-00205R1 

The budding yeast Fkh1 Forkhead associated (FHA) domain promotes the G1-chromatin state and activity of chromosomal DNA replication origins 

Dear Dr Fox, 

We are pleased to inform you that your manuscript entitled "The budding yeast Fkh1 Forkhead associated (FHA) domain promotes the G1-chromatin state and activity of chromosomal DNA replication origins" has been formally accepted for publication in PLOS Genetics! Your manuscript is now with our production department and you will be notified of the publication date in due course.

With kind regards,

Lilla Horvath

PLOS Genetics

On behalf of:
